



# Evaluating Microphysics and Boundary Layer Schemes in WRF: Assessment of 36 Scheme Combinations for 17 Major Storms in Saudi Arabia

Rajesh Kumar Sahu[1], Hamza Kunhu Bangalath[1], Suleiman Mostamandi[1], Jason Evans[2], Paul A. Kucera[3], and Hylke E. Beck[1]

[1]Physical Science and Engineering Division, King Abdullah University of Science and Technology (KAUST), Thuwal, Saudi Arabia
[2]Climate Change Research Centre, University of New South Wales, Sydney, Australia
[3]COMET Program, University Corporation for Atmospheric Research, Boulder, Colorado, USA

**Correspondence:** Hylke E. Beck (hylke.beck@kaust.edu.sa)

**Abstract.** Extreme rainfall events (EREs) and resulting flash floods in Saudi Arabia cause significant risks, including casualties and economic losses. Accurate simulations are crucial for forecasting, climate projections, and disaster management. This study evaluates boundary layer (BL) and cloud microphysics (MP) schemes to simulate EREs in the Arabian Peninsula (AP) using the Weather Research and Forecasting (WRF) model. Thirty-six combinations of four BL and nine MP schemes were

tested across 17 EREs at a convective-permitting 3-km resolution, compared with IMERG gridded satellite data for rainfall and station observations for temperature, humidity, and wind speed. Performance was assessed using Kling-Gupta Efficiency (KGE) incorporates correlation, variability, and overall bias. We found a good agreement between observed and simulated rainfall patterns, though some over- and underestimations were present. Among BL schemes, the Yonsei University (YSU) scheme stood out as the best performers in terms of rainfall, while Thompson (MP8) ranked the highest among the MP schemes.

Goddard (MP7) also delivered strong results. The Thompson-YSU combination yielded the highest mean KGE, performing statistically significantly better than 21 other combinations. Furthermore, performance rankings varied across meteorological variables, suggesting that superior rainfall performance does not necessarily correlate with an overall more accurate simulation. This study highlights the challenges of scheme evaluation and the importance of analyzing many EREs while using reliable reference data. It offers guidance for selecting the most appropriate schemes and lays the foundation for future ERE forecasting

and climate modeling improvements in arid regions.

## 1 Introduction

Extreme rainfall events (EREs) are episodes of intense precipitation over a short duration, often resulting in flash floods, landslides, and severe infrastructure damage (Easterling et al., 2000; Houze Jr, 2012; Kundzewicz et al., 2014; Srinivas et al., 2018). These events are becoming more frequent and intense as atmospheric moisture increases by about 7% per degree of

warming, following Clausius-Clapeyron scaling (e.g., Held and Soden, 2006; O'Gorman and Schneider, 2009). Although this rise in moisture is significant, mean precipitation increases at a slower rate of 2–3% per degree. In contrast, EREs can intensify





by as much as 6–10% (e.g., Allan and Soden, 2008; O'Gorman and Schneider, 2009), depending on their spatial and temporal scales, significantly increasing the potential for devastating impacts on vulnerable regions. Accurate prediction of EREs is critical for disaster planning, early warning systems, and water resource management, especially in places with a rising trend

of EREs (Luong et al., 2020; Attada et al., 2020).

Saudi Arabia, despite its arid desert climate and low annual precipitation, regularly experiences significant EREs (Almazroui, 2011; Haggag and El-Badry, 2013; Deng et al., 2015; Yesubabu et al., 2016; Almazroui et al., 2018; Atif et al., 2020; Attada et al., 2022) that often lead to dangerous flash floods, particularly during the rainy season from November to April. These events are often linked to the intrusion of intensified subtropical jet stream and mid-latitude cyclonic disturbances towards the

peninsula, combined with the low-level advection of warm, moist air from nearby water bodies, including the Red Sea, Persian Gulf, and Arabian Sea (Evans et al., 2004; Barth and Steinkohl, 2004; Evans and Smith, 2006; De Vries et al., 2013, 2016). The EREs, though infrequent, cause significant damage to infrastructure, agriculture, and communities. This makes accurate forecasting and projection of EREs critical for effective disaster management in the region. Reliable predictions can inform early warning systems, aid in the reanalysis of past events, and support climate change assessments. Given the increasing

frequency and intensity of EREs, numerical simulation plays a central role in the region's climate adaptation strategies, helping to improve early warning systems and inform long-term adaptation measures.

WRF is a widely used numerical model in the AP to simulate and forecast EREs (Deng et al., 2015; Almazroui et al., 2018; Taraphdar et al., 2021; Risanto et al., 2024). These models are subject to various sources of uncertainty, particularly due to parameterizations. Two key parameterization schemes include the Boundary Layer (BL) and cloud microphysics (MP)

schemes. The BL scheme governs the vertical exchange of momentum, heat, and moisture between the surface and the atmosphere, which is essential to simulate near-surface conditions. The MP scheme, on the other hand, controls cloud formation, precipitation processes, and interactions between water phases (Stull, 1988; Garratt, 1994; Stull, 2012; Dudhia, 2014). Accurate representation of the BL is particularly crucial for convective systems that often lead to EREs (Kumar et al., 2008). On the other hand, MP schemes simulate the evolution of cloud particles, including cloud droplets, rain, snow, and ice, which are

essential to determine the intensity and duration of rainfall (Dudhia, 2014). Although several previous studies have evaluated different schemes in AP (e.g. Deng et al., 2015; Schwitalla et al., 2020; Attada et al., 2022; Abida et al., 2022) and other arid and semi-arid regions (e.g. Zittis et al., 2014; Tian et al., 2017; Liu et al., 2021; Messmer et al., 2021; Khansalari et al., 2021; Mekawy et al., 2022; Pegahfar et al., 2022), they have typically focused on individual cases or performed limited sensitivity analyzes with a narrow range of parameterization schemes (Table 1). The case-specific nature of these studies often limits

the generalizability of the results across multiple events and varying conditions, which reduces their broader applicability for predicting EREs in the complex climate dynamics of the AP.

Our study addresses this gap by conducting an extensive evaluation of BL and MP schemes for simulating EREs at convection-permitting resolution (3 km) to establish the best combination of BL and MP parameterization schemes that consistently perform well across different EREs. We conduct sensitivity experiments on 17 ERE cases spanning from 2010 to 2022 across

the AP, testing 36 different combinations of BL and MP schemes to identify the optimal setup for ERE simulation in the AP.





We simulate the 17 extreme rainfall cases using a two-way nested domain configuration with 53 vertical levels and horizontal resolutions of 9 km and 3 km. To guide the reader, the paper is structured according to ten key questions:

1. Which BL scheme performs best in terms of rainfall?

2. Which MP scheme performs best in terms of rainfall?

3. Which component of the Kling-Gupta Efficiency (KGE) affects the final score the most?

4. How statistically significant are the differences in performance between scheme combinations in terms of rainfall?

5. How consistent are the temporal and spatial performance assessments for rainfall?

6. How consistent is the performance ranking among different variables?

7. What do the spatial patterns in simulated and observed rainfall look like for the events?

8. How well does the model perform in terms of the other variables?

9. Which BL and MP schemes were used in previous studies focusing on the Middle East?

10. How generalizable are our findings?

## 2 Physical geography and climatic description of the study area

Saudi Arabia, covering 80% of the AP, spans latitudes 16° to 33° N and longitudes 34° to 56° E, with an area of approximately 2,149,690 km², making it the largest country in the Middle East and the 12th largest globally. The terrain includes highlands, volcanic fields, mountain ranges, and the vast Arabian desert, featuring the Rub' al Khali, the world's largest continuous sand desert. Despite lacking permanent rivers, it has many wadis, alluvial deposits (Vincent, 2008; WeatherOnline, 2024), and about 1,300 islands in the Arabian Gulf and the Red Sea. The central plateau stretches from the Red Sea to the Arabian Gulf, while the Asir province reaches 3,002 meters at Jabal Ferwa, and the Hejaz region contains approximately 2,000 extinct volcanoes across 180,000 km². The climate is characterized by vast deserts, rugged mountains, and an arid climate (De Vries et al., 2016; El Kenawy et al., 2014; Mostamandi et al., 2022; Ukhov et al., 2020), with extreme summer temperatures of 45–54°C winters rarely below 0°C (Climate.com, 2018). Minimal rainfall except in the south, where monsoons bring around 300 mm of rain from October to March (Hasanean and Almazroui, 2015). The average annual rainfall over the region is about 114 mm (El Kenawy et al., 2014). The primary mechanisms driving precipitation vary between the eastern and western coasts. On the western coast, the Asir mountain chains play a significant role in capturing moist northwesterly winds along the Red Sea coast, particularly during winter, extending up to the Bab el-Mandeb Strait (Pedgley, 1974; El Kenawy et al., 2014; Mostamandi et al., 2022). From East Africa through the Red Sea towards the eastern Mediterranean, the Red Sea Trough (RST) creates a geographical environment conducive to forming strong low-pressure systems over the central Red Sea. These systems can





**Table 1.** Previous studies evaluating WRF physics schemes in the Middle East.

| Region | Kind of schemes | Number of Events | Conclusion | Reference |
|--------|-----------------|------------------|------------|-----------|
| Jeddah, Saudi Arabia | Microphysics schemes: Lin, Eta Ferrier | Three flash floods events | The WRF Model effectively simulates flash floods in Jeddah, with 1 km resolution improving rainfall accuracy and 5 km requiring careful parameterization due to observed spatial displacement. | Deng et al. (2015) |
| AP | Cumulus schemes: KF, BMJ, GF | Winter simulation from 2001 to 2016 | Selecting subgrid convective parameterization is crucial for accurate high-resolution rainfall simulations over the AP. | Attada et al. (2020) |
| AP | MP schemes: Thompson 2-moment, Thompson aerosol-aware and WDM6 and BL schemes: MYNN level 2.5 and YSU | Case study on July 14, 2015 | The best results were achieved using aerosol-aware Thompson microphysics with MYNN PBL, effectively capturing precipitation. | Schwitalla et al. (2020) |
| Middle East | BL schemes: ACM2, QNSE, MYNN | Single year run for 2017 | Gray-zone simulations enhance precipitation modeling but are highly dependent on resolution and the selection of physics schemes. | Taraphdar et al. (2021) |
| AP | Cumulus schemes: KF, BMJ, GF | Winter simulation from 2001 to 2016 | Extreme winter rainfall is best simulated using the KF scheme, highlighting the importance of cumulus parameterization in WRF for reliable modeling in the hyperarid AP region. | Attada et al. (2022) |

generate substantial rainfall within the region (De Vries et al., 2013; El Kenawy et al., 2014). In contrast, the eastern coast,
influenced by the Hajar Mountains and its proximity to the Arabian Sea, receives convective precipitation driven by the summer
monsoon and moisture-laden winds from the Indian Ocean (Babu et al., 2016).

## 3 Data and Methods

### 3.1 Selection of Historical Extreme Rainfall Cases

We selected 17 EREs that led to significant damage and casualties, resulting in widespread media attention. Table 2 lists the
EREs analyzed in this study. We considered 17 cases to increase our chance of obtaining statistically significant results. We



limited the number of cases to avoid excessive computational demand, as high-resolution simulations for each event require substantial processing power, storage and time.

## 3.2 Initial and Boundary Conditions

ERA5 reanalysis data (0.25° resolution; Hersbach et al., 2020) was utilized to provide initial and boundary conditions for
each 3-hour time step to run WRF-ARW. The ERA5 data was obtained via the Copernicus Climate Data Store (CDS; https://cds.climate.copernicus.eu) from the European Centre for Medium-Range Weather Forecasts (ECMWF). ERA5 is the most reliable reanalysis currently available and was used to derive the initial and boundary conditions.

## 3.3 Observations

As a reference for our assessment, we used rainfall estimates from the microwave-based Integrated Multi-satellite Retrievals
for GPM (IMERG) Final V07 (Huffman et al., 2023). The product covers 2000 to the present, has a 30-minute 0.1° resolution, and was aggregated hourly for our analysis. We also used radiosonde data to examine the vertical structures at different time steps (Supplement Figure S1). The 00:00 and 12:00 UTC radiosonde data were collected from the University of Wyoming (https://weather.uwyo.edu/upperair/sounding.html) for several stations, providing relative humidity (%), temperature (°C), and wind speed (m/s). Additionally, surface meteorological information (Supplement Figure S2), including 2-m temperature (°C),
relative humidity (%), and wind speed (m/s), was obtained from the IOWA Environmental Mesonet data provided by Iowa State University (https://mesonet.agron.iastate.edu/request/download.phtml?network=SA__ASOS).

## 3.4 WRF Model Configuration

This study uses the Advanced Research version of Weather Research and Forecasting (WRF-ARW) model version 4.4, a non-hydrostatic, fully compressible model with a terrain-following coordinate system (Skamarock et al., 2019). The model
is configured with two-way nested domains, with 53 vertical hybrid sigma levels and a horizontal resolution of 3 km in the innermost domain, as shown in Figure 1. The AP domain covers a vast region from 21°E to 65°E in the zonal direction and from 2°N to 40°N in the meridional direction, allowing for the representation of large-scale atmospheric features and internal dynamics. The study performed 612 distinct simulations, each with a specific BL and MP scheme combination, to thoroughly evaluate the combined performance of these configurations across 17 EREs. The WRF model used in this study explicitly
simulates convection within the inner domain at a 3 km resolution (convection-permitting). In contrast, the outer domain relies on a convection parameterization scheme (Snook et al., 2019).

We considered 36 combinations involving nine MP and four BL schemes. The BL schemes tested include Mellor-Yamada Nakanishi Niino (MYNN) Level 2.5 and Level 3 (BL5, BL6; Nakanishi and Niino, 2006), Yonsei University (YSU; BL1; Hong et al., 2006), and Bougeault-Lacarrère (BouLac; BL8; Bougeault and Lacarrere, 1989), while the MP schemes include
Kessler (MP1; Kessler, 1969), Purdue Lin (MP2; Chen and Sun, 2002), WRF Single-Moment 3-class and 5-class (MP3 and MP4, respectively; (Hong et al., 2004)), Eta Ferrier, (MP5; Rogers et al., 2001), WRF Single-Moment 6-class (MP6; Hong



and Lim, 2006), Goddard (MP7; Tao et al., 2016), Thompson (MP8; Thompson et al., 2008), and Morrison 2-Moment (MP10; Morrison et al., 2009). These combinations were selected based on their compatibility with the surface layer physics Revised MM5 scheme (Jiménez et al., 2012), and additional schemes were not included due to the higher computational and storage

demands. Previous studies focusing on the AP have also utilized these schemes, including (Deng et al., 2015; Attada et al., 2022; Luong et al., 2020; Schwitalla et al., 2020). Each combination was examined for its ability to reproduce severe rainfall occurrences in various Saudi Arabian areas correctly.

Initial and boundary conditions were extracted from ERA5 reanalysis data at 3-hour intervals with a 0.25° resolution. The model simulations were run for 84 hours, including a 48-hour spin-up, with the analysis focused on the 24-hour event window

for each ERE. Refer to Table 2 for the simulation start dates and Table 3 for the model configuration.

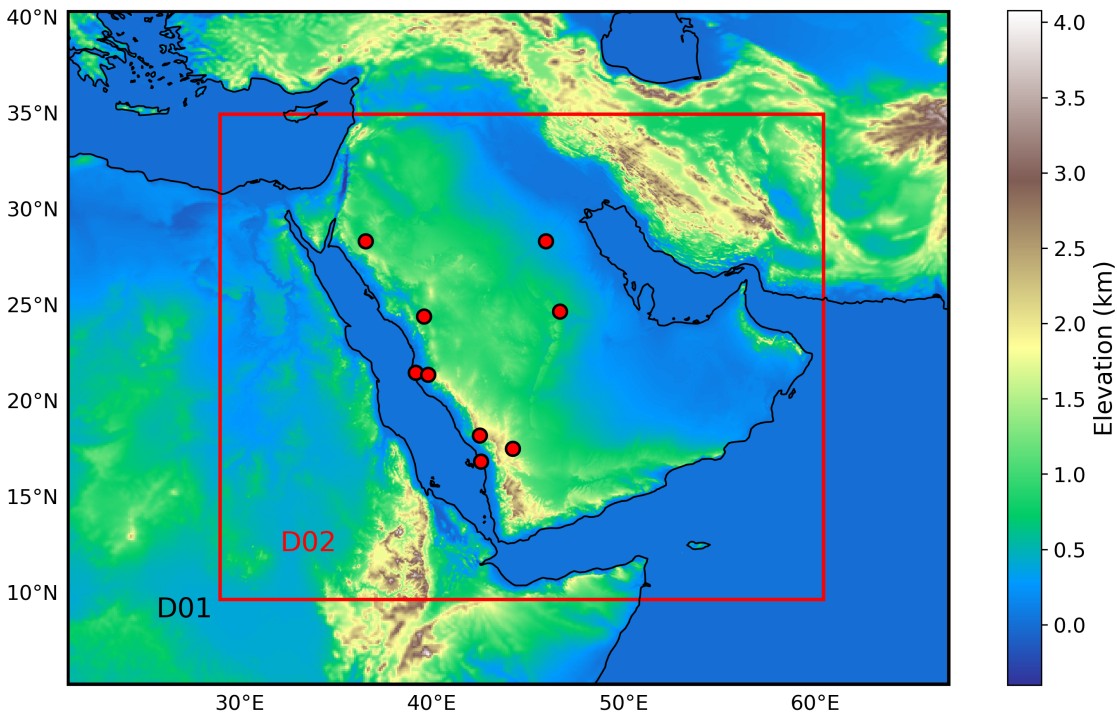

**Figure 1.** WRF-ARW domain for the AP region showing the elevation in the background and radiosonde locations as red markers.

## 3.5 Model Assessment Approach

Each combination of MP and BL schemes was extensively evaluated using the Kling-Gupta Efficiency (KGE; Gupta et al., 2009; Kling et al., 2012). KGE is a metric used as a comprehensive measure that analyzes correlation, bias, and variability between simulated and observed data. For rainfall, the KGE was calculated separately in space and in time. For the temporal

KGE, we first calculated, for each hour of the event day (refer to Table 2), the spatial average of observed and simulated



**Table 2.** Extreme rainfall events in the Arabian Peninsula selected to determine the efficacy of different MP and BL scheme combinations. Abbreviations: N=north, E=east, S=south, W=west, and P=people.

| Event Date | Location | Simulation Start | Rainfall | Fatalities / Impact | Source |
|---|---|---|---|---|---|
| 24-11-2022 | Jeddah, Makkah, and western Saudi Arabia (W) | 22-11-2022-00:00 | 179 mm | 2 P died in flooding | FloodList (www.floodlist.com) |
| 27-04-2021 | Makkah (W) | 25-04-2021-00:00 | Unknown | Severe flooding reported | FloodList |
| 04-02-2021 | Tabuk (NW), Hafr Al-Batin (E) | 02-02-2021-00:00 | 43 mm in 30 min | 7 P died; 1,100 P affected | General Directorate of Civil Defense (CDD) |
| 27-10-2019 | Hafr Al-Batin (E) | 25-10-2019-00:00 | 43 mm in 30 min | 7 P died, 11 P injured; 1,100 P affected | FloodList |
| 23-05-2019 | Jazan, Najran (SW) | 21-05-2019-00:00 | Unknown | 1 P missing in floods | FloodList |
| 08-02-2019 | Madinah (W), Tabuk (NW), Riyadh (E), others | 06-02-2019-00:00 | 36.6 mm in 24 hrs | 4 P died; many rescued | FloodList |
| 28-01-2019 | Tabuk (NW), Riyadh (C), Jeddah (W), others | 26-01-2019-00:00 | Unknown | 1 P died; 30 P evacuated | CDD |
| 20-11-2017 | Jeddah, Hail (W) | 18-11-2017-00:00 | 115.5 mm/hr | 4 P died; 481 rescued | FloodList |
| 14-02-2017 | Asir (SW), Dammam (E), others | 12-02-2017-00:00 | 90 mm in 24 hrs | 1 P died; 10 P injured | CDD |
| 28-11-2016 | Asir (SW), Riyadh (C), others | 26-11-2016-00:00 | Unknown | 8 P died; 120 evacuated | FloodList |
| 08-04-2016 | Asir, Baha, Taif (S) | 06-04-2016-00:00 | Unknown | 3 P died in Al-Baidhani valley | FloodList |
| 24-11-2015 | Riyadh, Al-Qassim | 22-11-2015-00:00 | Unknown | 1P died | FloodList |
| 28-10-2015 | Northern Saudi Arabia (N) | 26-10-2015-00:00 | Unknown | 6 P died | FloodList |
| 23-03-2015 | Riyadh (C), Al Bahah (NW) | 21-03-2015-00:00 | Unknown | 11 P died; 300 P rescued | FloodList |
| 20-11-2013 | Riyadh (C), Arar (N) | 18-11-2013-00:00 | Unknown | 4 P died | CDD |
| 14-01-2011 | Jeddah (W) | 12-01-2011-00:00 | 110 mm in 3 hrs | 10 P died; 1,500 P missing | CDD |
| 30-12-2010 | Jeddah (W) | 28-12-2010-00:00 | Unknown | No data | CDD |

rainfall across Domain 2 (D02; Figure 1). The KGE was derived from these 24 pairs of observed and simulated spatially




**Table 3.** WRF-ARW (Version 4.4) model configuration used in this study.

| Configuration Parameter | Details |
| --- | --- |
| Dynamics | Non-hydrostatics |
| Boundary and initial conditions | ERA5 reanalysis |
| Data Interval | 3 hours |
| Resolution | D01 9 km and d02 3 km |
| Map Projection | Mercator |
| Horizontal grid system | Arakawa-C grid |
| Integration time step | 30 s |
| Vertical coordinates | Terrain-following hydrostatic pressure vertical coordinate with 53 vertical levels |
| Time integration scheme | 3rd-order Runge-Kutta Scheme |
| Spatial differencing scheme | 6th-order centre differencing |
| Microphysics Parameterization (MP) | Kessler, Purdue Lin, WRF Single-moment 3-class (WSM3), WRF Single-moment 5-class (WSM5), Eta (Ferrier), WRF Single-moment 6-class (WSM6), Goddard, Thompson graupel, Morrison 2–moment |
| Cumulus Parameterization (CU) | D01 (Kain Fritsch), D02 (no physics used) |
| Planetary Boundary Layer (BL) Parameterization | Yonsei University Scheme (YSU), Mellor-Yamada Nakanishi and Niino Level 2.5, Mellor-Yamada Nakanishi and Niino Level 3, BouLac |
| Surface layer parameterization | Noah Land Surface Scheme |
| Surface Layer Physics | Revised MM5 (Jiménez et al., 2012) |
| Short wave radiation (ra_sw_physics) | RRTMG scheme (Iacono et al., 2008) |
| Long wave radiation (ra_lw_physics) | RRTMG scheme |

averaged values. For the spatial KGE, for each grid cell within D02, the daily mean of observed and simulated rainfall was computed. The KGE was subsequently calculated using these observed and simulated grid cell daily means. The formula for KGE is given by:

$$\text{KGE} = 1 - \sqrt{(r-1)^2 + (\beta-1)^2 + (\gamma-1)^2}, \tag{1}$$

where $r$ is Pearson's correlation coefficient between the observed and simulated data, $\beta$ is the ratio of the mean simulated data to the mean observed data, assessing the bias, and $\gamma$ is the ratio of the coefficient of variation of simulated data to that of the coefficient of variation of observed data, evaluating the variability.

For 2-meter temperature, 2-meter relative humidity, and wind speed, KGE was calculated from hourly METAR observations from the IOWA Mesonet and corresponding simulations from the nearest model grid-cell for the day of each event.



## 4 Results and Discussion

### 4.1 Which BL scheme performs best in terms of rainfall?

The selection of an appropriate BL scheme is crucial for accurately simulating extreme rainfall in subtropical desert regions, such as Saudi Arabia, due to unique environmental factors (e.g., Taraphdar et al., 2021). Intense surface heating in deserts
leads to the development of extremely deep BLs, reaching up to 5 km during the day (e.g., Gamo, 1996; Marsham et al., 2008; Ntoumos et al., 2023). This necessitates a scheme capable of accurately modeling the vertical distribution of heat, moisture, and momentum in a deeper BL. Complex thermodynamic profiles, with sharp temperature gradients and significant humidity variations, further complicate modeling. Accurately capturing these conditions is essential for simulating extreme rainfall. Deserts also experience strong diurnal temperature variations, necessitating a BL scheme that effectively handles short-term
fluctuations in energy and moisture fluxes between the surface and the atmosphere (e.g., Taraphdar et al., 2021).

Rainfall in the AP typically occurs during the winter months. It is driven by the interaction of frontal systems, formed between cold, dry extratropical air and hot, moist air from nearby seas (e.g., Taraphdar et al., 2021). EREs are frequently linked to mesoscale convective systems (MCS), initiated by either frontal passages or orographic lifting in mountainous regions (e.g., De Vries et al., 2016; El Kenawy and McCabe, 2016; Yesubabu et al., 2016; Luong et al., 2020). These convective systems rely
heavily on properly representing turbulence and mixing within the BL. Desert regions are also characterized by rapid changes in atmospheric conditions over short time scales, which requires the use of advanced BL schemes.

Figure 2 presents the temporal KGE scores in 36 combinations of BL-MP and the 17 EREs. The mean temporal and spatial KGE for the BL schemes—YSU (BL1), MYNN Level 2.5 (BL5), MYNN Level 3 (BL6), and BouLac (BL8)—are summarized in Table 4. Among these, the YSU (BL1) scheme showed superior performance among the BL schemes. Notably, YSU (BL1) is
the only scheme with a non-local approach, unlike the other schemes, which are all local. This non-local mixing likely explains YSU's superior performance, enabling enhanced vertical mixing across the entire BL. Non-local schemes like YSU (BL1) can represent large eddy structures and transport heat, moisture, and momentum over considerable vertical distances, a capability that is particularly crucial in arid environments with intense surface heating and sharp thermal gradients, such as Saudi Arabia (Hong et al., 2006; Hu et al., 2010). In contrast, local schemes like the MYNN Level 2.5 (BL5), MYNN Level 3 (BL6) and
BouLac (BL8) rely on gradients at specific vertical levels and small-scale turbulence, which restricts their ability to simulate deep convection and rapid vertical mixing (Nakanishi and Niino, 2006; Bougeault and Lacarrere, 1989).

Previous research has shown that non-local schemes, including YSU (BL1), yield a deeper and more accurately structured BL than local schemes, especially in the presence of strong surface heating and convective activity, which are characteristic of desert climates (Xie et al., 2012; Cohen et al., 2015). Specifically, YSU's non-local treatment of BL processes allows it to
develop a deeper BL during the daytime, a typical feature in arid regions, enhancing the scheme's ability to capture severe convective activity (Cohen et al., 2015).

The YSU scheme's (BL1) performance in representing BL processes is especially advantageous in regions where convection is often triggered by advancing frontal systems, as is common in the AP. In a case study using the WRF model, Cohen et al. (2015) demonstrated that YSU's non-local treatment improves the BL's response to cold fronts, triggering convection more





realistically and enhancing features like the formation of double lines of intense convection. This improvement arises because YSU (BL1) minimizes the dilution of moist air by dry air entrainment, maintaining a higher moisture concentration within the BL. This "fuel" is crucial for sustaining severe convection when fronts initiate it, particularly in desert regions, where dry air entrainment can otherwise weaken or inhibit intense convective activity and thus reduce the accuracy of ERE simulations.

In contrast, local schemes like MYNN (BL5 and 6) and BouLac (BL8) are optimized for stable or stratified BLs, performing well by simulating small-scale turbulence. However, these schemes often struggle in unstable, highly convective environments like those in Saudi Arabia, where larger eddy structures dominate and require extensive vertical mixing to capture intense updrafts and precipitation (Hu et al., 2013; Cohen et al., 2015). Therefore, YSU's non-local approach, with its integrated vertical mixing and responsiveness to strong surface heating likely contributed to its superior performance in simulating EREs, capturing the necessary BL transitions and intense convective plumes critical for accurate ERE representation in the desert regions of Saudi Arabia.

Performance is consistently lower for the BL6 scheme (Mellor-Yamada Nakanishi Niino Level 3; mean KGE of 0.26; Figure 2) scheme, and it consistently showed lower and sometimes negative KGE scores across different MP schemes. The scheme's higher-order local closure approach can lead to over-diffusion, dampening essential vertical motions and limiting its ability to capture coherent eddies and large-scale vertical transport—critical for effective moisture and heat distribution needed for convective rainfall (Nakanishi and Niino, 2006; Shin and Hong, 2011). Meanwhile, the BL8 (Bougeault-Lacarrère) and BL5 (Mellor-Yamada Nakanishi Niino Level 2.5 — MYNN) schemes (mean KGE of 0.41 and 0.38, respectively; Figure 2) also show reasonable but lower performance than YSU (BL1), indicating that their local turbulence closures may similarly restrict effective representation of key atmospheric dynamics, particularly in arid environments where accurate BL processes are essential (Hu et al., 2010).

## 4.2 Which MP scheme performs best in terms of rainfall?

Figure 2 presents temporal KGE scores across 36 BL-MP combinations and the 17 EREs. The mean temporal and spatial KGE for various MP schemes, including Kessler (MP1), Purdue Lin (MP2), WSM3 (MP3), WSM5 (MP4), Eta Ferrier (MP5), WSM6 (MP6), Goddard (MP7), Thompson (MP8), and Morrison (MP10), are presented in Table 4. The Goddard (MP7) and Thompson (MP8) schemes achieved the highest mean KGE scores. This is likely due to their sophisticated handling of cloud MP, especially in representing mixed-phase and ice-phase processes essential for simulating EREs in arid regions like Saudi Arabia. Though Goddard is a single-moment scheme, it includes detailed processes for ice, snow, and graupel, making it effective for capturing intense convective storms driven by complex thermodynamics and rapid cloud development (Tao, 2003). Its optimized treatment of rain formation and melting allows it to handle the rapid updrafts and temperature variations characteristic of desert climates, where efficient particle formation and fallout are crucial for high-intensity rainfall events.

As a double-moment approach, the Thompson scheme (MP8) further enhances these capabilities by dynamically adjusting particle size distributions, including cloud droplets, rain, ice and snow. This adaptability allows it to respond effectively to environmental changes typical of desert frontal systems, where intense updrafts can quickly alter particle sizes (Thompson et al., 2008). The double-moment structure offers flexibility in tracking a broad range of particle sizes, enabling Thompson





**Table 4.** Mean KGE values for temporal and spatial assessments of MP and BL schemes.

| Scheme | Temporal KGE | Spatial KGE |
|---|---|---|
| **MP Schemes** | | |
| Kessler (MP1) | 0.26 | 0.05 |
| Purdue Lin (MP2) | 0.35 | 0.27 |
| WRF Single-Moment 3-class (WSM3; MP3) | 0.41 | 0.30 |
| WRF Single-Moment 5-class (WSM5; MP4) | 0.39 | 0.25 |
| Eta Ferrier (MP5) | 0.39 | 0.28 |
| WRF Single-Moment 6-class (WSM6; MP6) | 0.36 | 0.28 |
| Goddard (MP7) | 0.42 | 0.33 |
| Thompson (MP8) | 0.42 | 0.31 |
| Morrison (MP10) | 0.30 | 0.29 |
| **BL Schemes** | | |
| YSU (BL1) | 0.43 | 0.29 |
| Mellor-Yamada Nakanishi Niino Level 2.5 (MYNN; BL5) | 0.38 | 0.27 |
| Mellor-Yamada Nakanishi Niino Level 3 (MYNN: BL6) | 0.26 | 0.21 |
| Boulac (BL8) | 0.41 | 0.27 |

to simulate light and heavy precipitation effectively. This capability is crucial in arid regions, where rapid shifts between

intense precipitation and dry conditions are common, and tracking both mass and concentration enhances the accuracy of these

transitions.

The superior performance of these schemes over simpler single-moment models, like Kessler (MP1), Purdue Lin (MP2), or WSM3 schemes (MP3), underscores the importance of advanced microphysical processes for capturing ERE variability and intensity. Simpler schemes lack adaptability to evolving particle size distributions, limiting their effectiveness in highly

convective environments with rapid shifts. Notably, despite its advanced double-moment structure, Morrison underperformed, possibly due to sensitivities that may hinder accuracy in arid, convective conditions—a point warranting further study beyond this scope. These results highlight the importance of selecting MP schemes with detailed ice and mixed-phase processes when modeling EREs in desert regions.

## 4.3 Which component of the Kling-Gupta Efficiency (KGE) affects the final score the most?

Figure 3a presents the values of KGE and its components — correlation, bias, and variability ($r$, $\gamma$, and $\beta$, respectively; Eq. 1) — for all 17 events for the best performing Thompson-YSU scheme (MP8_BL1) for rainfall. In the interest of conciseness, we focus only on the temporal KGE results here, as the spatial KGE results are quite consistent (see Section 4.1 and 4.2 and Table 4).





**Figure 2.** Temporal KGE scores for precipitation of 36 schemes combined for 17 EREs.

Correlation is sensitive to the timing of events, variability ratio is sensitive to the distribution, and bias reflects the mean. For the best scheme (Thompson-YSU; MP8_BL1), the mean KGE score for precipitation across 17 events is 0.48. Decomposing this score into the three components, expressed as $|r - 1|$, $|\beta - 1|$ and $|\gamma - 1|$ to make them comparable, yields mean values of 0.33, 0.23, 0.25, respectively, where values closer to 0 indicate better performance. Among the three KGE components, the scheme thus performed worst in terms of correlation, and this subcomponent thus exerted the dominant influence on the



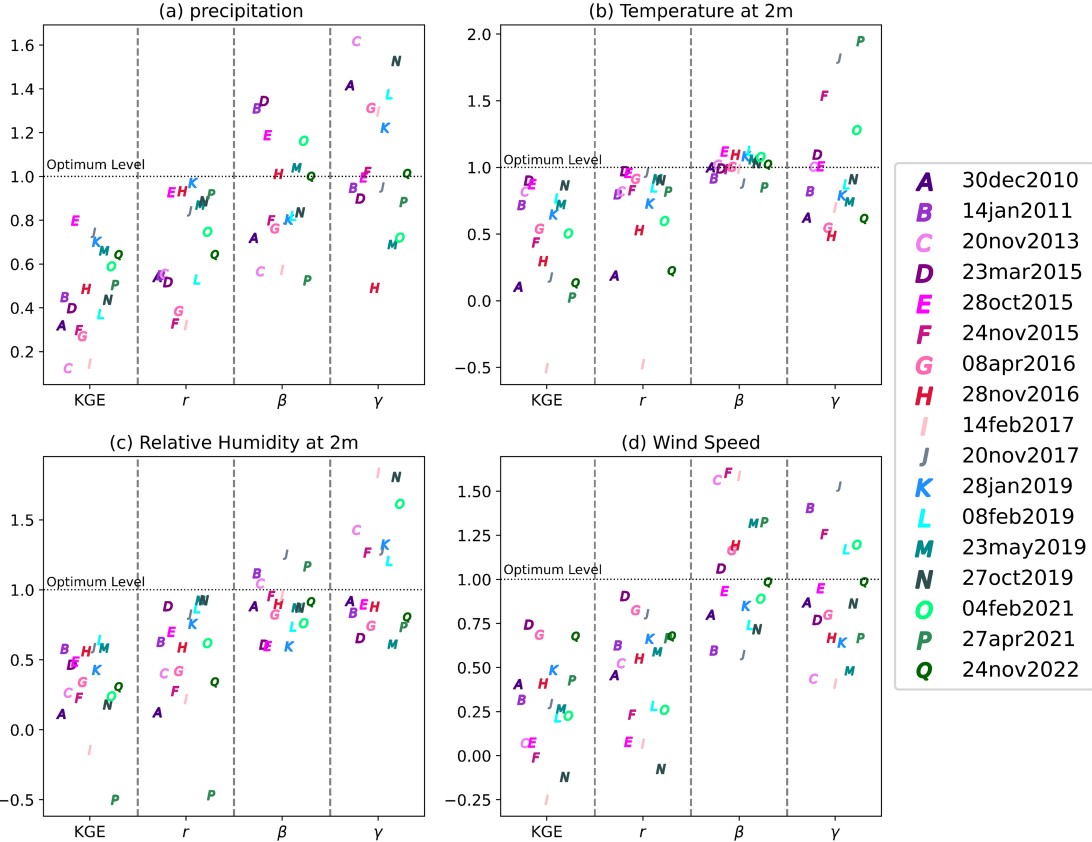

**Figure 3.** Correlation coefficient ($r$), long-term bias ($\beta$), and variability ratio ($\gamma$) used to calculate the KGE values for the best-performing scheme across 17 EREs for (a) precipitation, (b) 2-m temperature, (c) 2-m relative humidity, and (d) wind speed. The letters (A, B, ...,Q) indicate the 17 different EREs.

final KGE scores. This suggests that in order to get an improved KGE score, the most important component score to improve

is the correlation, which, in our evaluation, is related to the timing of events. The mean KGE value across all other schemes and events is 0.36, and the mean values for $|r-1|$, $|\beta-1|$ and $|\gamma-1|$ are 0.34, 0.29, and 0.24, respectively. This suggests that the correlation also tends to exert the dominant influence for the other scheme combinations, while bias also plays a role. The mean KGE score for the worst-performing scheme combination — Morrison-MYNN (MP10_BL6) — is 0.13, while the mean values of the three KGE components $|r-1|$, $|\beta-1|$ and $|\gamma-1|$ are 0.33, 0.57, and 0.36, respectively. This scheme thus

performs particularly poorly in terms of bias.




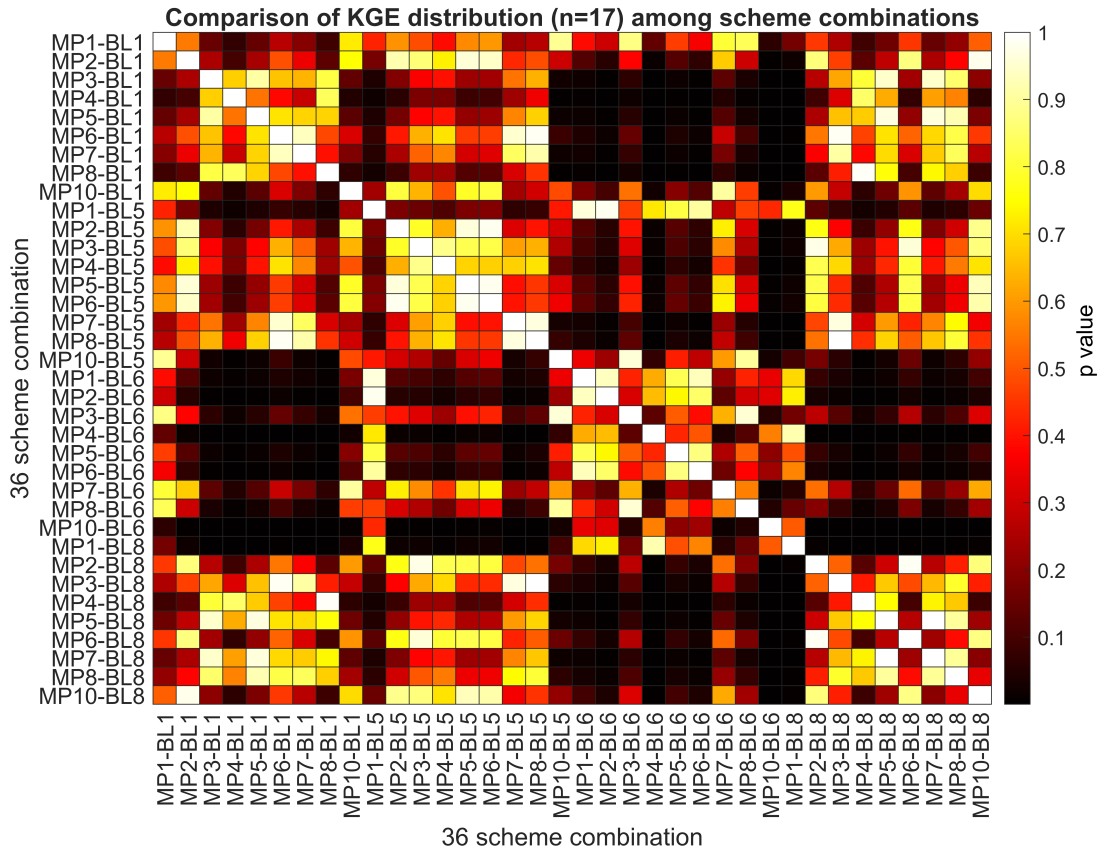

**Figure 4.** Pairwise p-values from independent t-tests comparing the ΔKGE distributions of 36 scheme combinations for rainfall. ΔKGE values were calculated by subtracting the mean KGE across events from the KGE values presented in Figure 2. A p-value threshold of 0.1 was used to identify statistically significant differences between scheme combinations.

## 4.4 How statistically significant are the differences in performance between scheme combinations in terms of rainfall?

The differences in KGE between different scheme combinations are generally relatively small. For example, the best-performing scheme combination (Thompson-YSU; MP8_BL1) achieved a mean KGE of 0.48, while the second best-performing scheme combination (Goddard-YSU; MP7_BL1) achieved a mean KGE of 0.44 (Figure 2). Furthermore, the corresponding standard deviations across events are 0.20 and 0.24, respectively, indicating substantial variability in scores among events. Additionally, the consistency in performance ranking among events is fairly low (Figure 5). This raises the question of whether the observed differences in performance between scheme combinations are statistically significant and, hence, whether our evaluation approach is adequate for determining the relative performance of different scheme combinations, which is the primary objective of this study.





To address this question in the context of rainfall, we calculated ΔKGE scores by subtracting the mean KGE across events from the KGE values presented in Figure 2 to eliminate systematic differences in scores among events. We then tested whether the distributions of ΔKGE values for different scheme combinations are statistically similar or different using pairwise independent t-tests. Figure 4 presents a 36x36 matrix of p-values, which reveals that the best-performing scheme combination

(Thompson-YSU; MP8_BL1) significantly outperformed 21 other scheme combinations (at a p-level of 0.1), whereas the worst-performing scheme combination (Morrison-MYNN; MP10_BL6) performed significantly worse than 28 other scheme combinations (also at a p-level of 0.1). These results confirm that our assessment provides meaningful and statistically significant insights into the relative performance of different scheme combinations. However, our assessment does not definitively identify a single best-performing scheme but instead highlights groups of better- and worse-performing schemes.

We further analyzed the spatial variation of ΔKGE, as illustrated in Supplement Figure S4. The 36×36 p-value matrix provides a statistical comparison of scheme combinations, highlighting their relative performance. The results indicate that the best-performing scheme combination (Thompson-YSU; MP8_BL1) exhibits significantly higher skill than 21 other scheme combinations at a statistical significance p level of 0.1. Similarly, the worst-performing scheme combination (Morrison-MYNN; MP10_BL6) demonstrates significantly lower skill than 28 other scheme combinations at the same significance p

level 0.1. These findings align with the temporal variation of delta KGE analysis, reaffirming the robustness of Thompson-YSU (MP8_BL1) and the limitations of Morrison-MYNN (MP10_BL6) in accurately simulating precipitation dynamics.

The ability of an assessment such as this to detect significant differences in performance between schemes depends on the mean and standard deviation of the ΔKGE distribution. Assuming a standard deviation of 0.15 (equivalent to that of Thompson-YSU; MP8_BL1), the current sample size of 17 events requires a mean ΔKGE difference greater than 0.06 between schemes

to yield a statistically significant difference at a p-level of 0.1. Analyzing a larger sample of events would reduce the required mean difference, making it easier to detect significant differences in performance between schemes. For example, if we were to analyze 50 events, the required difference in mean ΔKGE would be just 0.03 (assuming again a standard deviation of 0.15). However, analyzing a larger number of events is computationally more expensive.

The standard deviation (i.e., the variability in ΔKGE among events) and, hence the number of events required to detect

significant performance differences between schemes is partly influenced by the quality of the reference data. In this study, we used a satellite-based precipitation dataset (IMERG-Final V07), which was associated with greater uncertainty than other reference datasets, such as radar data (Beck et al., 2019). This increased uncertainty may have contributed to higher variability in KGE scores (Evans and Imran, 2024). Unfortunately, radar data are not available in Saudi Arabia. Due to the strong correlation between different datasets and the fact that IMERG-Final significantly outperforms other datasets (Wang et al., 2025a),

we were unable to quantify the uncertainty arising from the choice of reference data.

### 4.5 How consistent are the temporal and spatial performance assessments for rainfall?

We calculated KGE scores temporally and spatially to assess the performance of the 36 BL-MP scheme combinations across the 17 EREs. The temporal KGE results for rainfall are presented in Figure 2, while the spatial KGE results for rainfall are provided in Supplement Figure S3. The mean KGE values categorized by MP and BL schemes, for both temporal and spatial




assessments, are summarized in Table 4. The overall mean temporal KGE across all schemes and events for rainfall is 0.37, whereas the overall mean spatial KGE is 0.26. This indicates that the simulations are more effective at capturing temporal variations in rainfall than spatial variations. This is expected, as rainfall in the region is highly localized, and models often struggle to replicate the spatial distribution of events precisely. Overall, we found a strong consistency in the overall ranking of schemes between the temporal and spatial assessments, with a Spearman rank correlation of 0.65 (p-value of 0.00) between the

mean temporal and spatial KGE values for the scheme combinations. In both the temporal and spatial assessments, the Goddard (MP7) and Thompson (MP8) MP schemes, particularly when paired with the YSU (BL1) BL scheme, consistently emerged as superior. Conversely, the Kessler (MP1) scheme with MYNN level 3 (BL6) scheme performed worst in both assessments.

### 4.6   How consistent is the performance ranking among different variables?

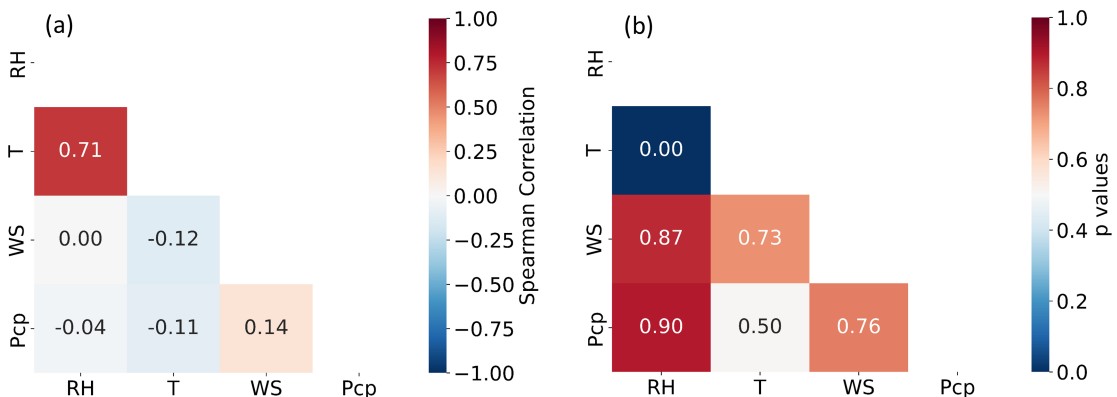

**Figure 5.** (a) Mean Spearman correlation coefficients and (b) corresponding median p values calculated among mean KGE scores for different meteorological variables, indicating the degree of consistency in performance rankings among variables. Variable definitions: relative humidity = RH; temperature = T; wind speed = WS; and precipitation = Pcp.

    Ideally, if our conclusions about the performance of various MP and BL scheme combinations regarding rainfall are valid,

and if this superior performance truly reflects a model that better represents reality (i.e., we are 'getting the right results for the right reasons'), then the performance ranking for rainfall should align with those of other variables. To investigate this, we calculated Spearman rank correlations and corresponding $p$-values between the temporal mean KGE scores for different variables (Figure 5) indicating the degree of consistency in performance rankings among these variables. The meteorological variables considered were relative humidity, temperature, wind speed, and precipitation.

Most variable pairs exhibited insignificant correlations except for temperature and relative humidity, which are intrinsically linked through the Clausius-Clapeyron relationship as temperature controls saturation vapor pressure and, thus, relative humidity. The lack of significant correlations might have three potential explanations. First, although we considered the possibility of unreliable reference data causing discrepancies in model performance, the robustness of our reference datasets — IMERG for precipitation, radiosonde, and IOWA Environmental Mesonet data for other variables — makes this explanation





less likely. Second, although MP and BL schemes strongly influence precipitation simulation, other model components like land surface schemes, which affect soil moisture and heat fluxes, and radiation schemes, which affect surface and atmospheric energy balances, may have a more pronounced impact on variables such as temperature and wind speed. Third, there might be compensatory behavior within the model, where improvements in simulating one variable do not necessarily result in a more realistic simulation and may yield reduced performance in others. This phenomenon, where models achieve the right results

for the wrong reasons, is not uncommon in geosciences and poses significant challenges in model evaluation and improvement (Kirchner, 2006; Parker, 2006; Knutti, 2010; Hourdin et al., 2017; Broecker, 2017; Krantz et al., 2021).

Studies have shown that the choice of parameterization schemes significantly affects model performance across different variables and regions. For example, a high-resolution regional climate model physics ensemble over Europe demonstrated that optimal configurations vary depending on the specific climate variable and region under consideration (Laux et al., 2021). WRF

model has indicated that its performance is highly sensitive to the selection of physical parameterization schemes, particularly in regions with complex terrain and variable climates (Pervin and Gan, 2021). Therefore, a more detailed analysis of the model's performance in simulating the various processes contributing to rainfall in each case is necessary. While this is beyond the scope of the current paper, the authors intend to explore these questions in future research.

### 4.7 What do the spatial patterns in simulated and observed rainfall look like for the events?

Figures 6 and 7, respectively, present observed (IMERG-Final V07) and simulated (WRF) 24-hr rainfall accumulations for the 17 selected rainfall events. The WRF simulations were generated using the best-performing scheme (Thompson-YSU; MP8_BL1). Overall, WRF generally seems to capture reasonably well the location, extent, and amounts indicated by IMERG. For example, the strong convective systems with high-intensity localized rainfall exceeding 120 mm on events like 20-Nov-2013 and 28-Jan-2019 are captured well. However, the model overestimates rainfall in several events (e.g., 08-Feb-2019) and

underestimates rainfall in others (e.g., 28-Oct-2015). While WRF generally captures the broad patterns, the lack of a better match is attributable to several reasons. First, potential deficiencies in the MP, BL and convection schemes and other model simplifications lead to potential inaccuracies in moisture convergence and convective updrafts (Taraphdar et al., 2021; Attada et al., 2022). Second, we used ERA5 as boundary conditions to force the model, and while ERA5 is the best reanalysis currently available, it nonetheless is subject to random errors and bias (Hersbach et al., 2020; Soci et al., 2024). Third, we did not include

data assimilation or nudging (Lei and Hacker, 2015; Feng et al., 2021), two important techniques to improve the simulations. Fourth and finally, the IMERG data, though found to perform relatively well in precipitation product evaluations (Abbas et al., 2025; Wang et al., 2025b), nonetheless carries significant uncertainty in the region.

### 4.8 How well does the model perform in terms of the other variables?

While the previous subsections primarily focused on rainfall, it is worthwhile to investigate how the model performs in terms

of other meteorological variables. To this end we analyzed the KGE components for T, RH, and WS as presented in Figure 3b to 3d. Figure 3b presents values of KGE and its components ($r$, $\gamma$, and $\beta$; Eq. 1; Gupta et al., 2009; Kling et al., 2012) for all 17 events for the best performing scheme (MP8_BL1) for T. For the best scheme, the mean KGE score for T across 17







**Figure 6.** Daily accumulated rainfall from our observation-based data source (IMERG-Final V07) for the 17 extreme events.

events is 0.47, while the mean scores for $|r-1|$, $|\beta-1|$ and $|\gamma-1|$ are 0.32, 0.06, 0.33, respectively. Among the three KGE components, the scheme thus performed worst in terms of correlation and variability, and these two components thus exert the dominant influence on the final KGE scores. This suggests that to get an improved KGE score of T, the most important component scores to improve are correlation and variability. If we look at mean KGE values across all other schemes across 17 EREs is 0.45, and the mean scores for $|r-1|$, $|\beta-1|$ and $|\gamma-1|$ are 0.35, 0.06, 0.34, respectively. This suggests that, for all other scheme combinations, the correlation and variability components dominate the KGE values.



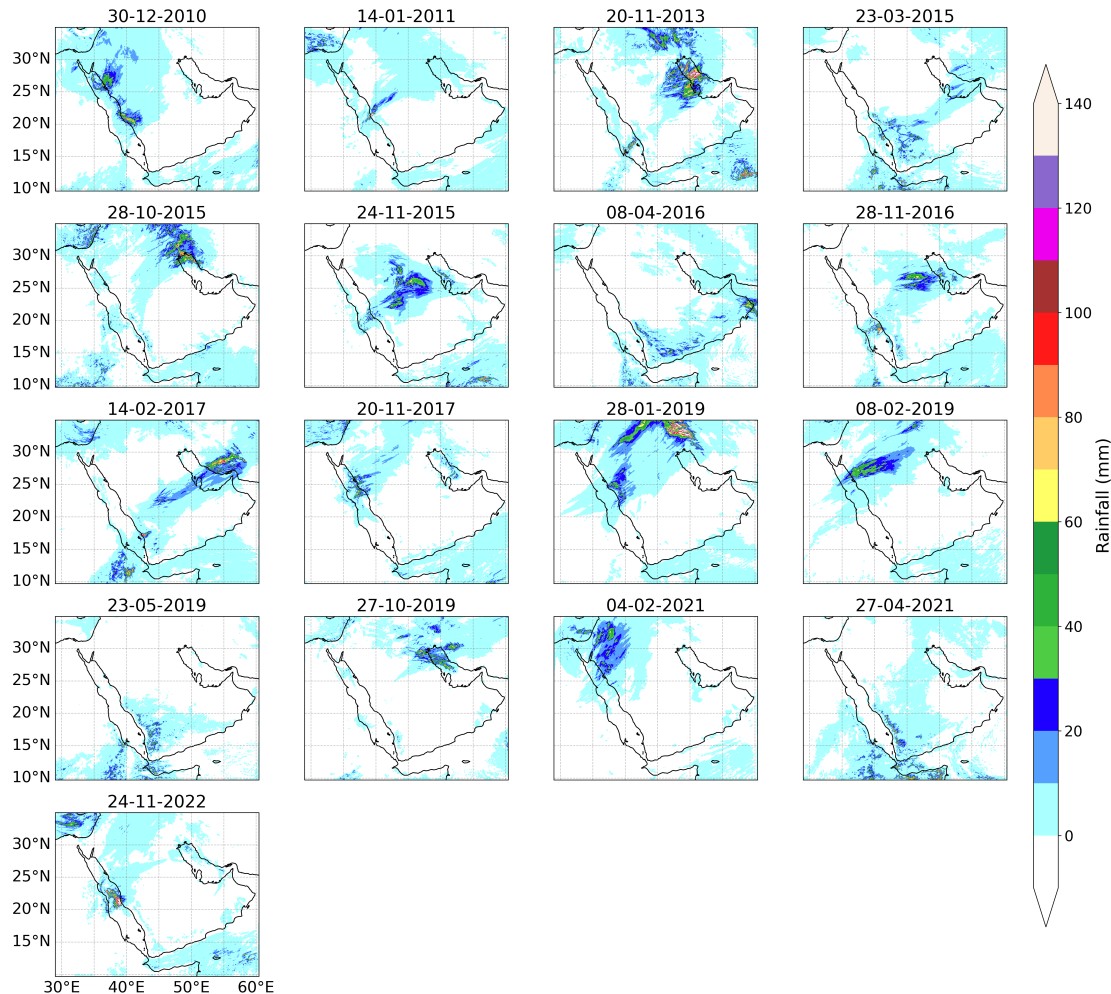

**Figure 7.** Daily accumulated rainfall from WRF using the performing scheme combination (Thompson-YSU; MP8_BL1) for the 17 extreme events.

Figure 3c presents the values of KGE and its components ($r$, $\gamma$, and $\beta$; Eq. 1; Gupta et al., 2009; Kling et al., 2012) for all 17

events for the best performing scheme (Thompson-YSU; MP8_BL1) for RH. For the best scheme, the mean KGE score for RH across 17 events is 0.31, while the mean scores for $|r-1|$, $|\beta-1|$ and $|\gamma-1|$ are 0.47, 0.18, 0.33, respectively. Among the three KGE components, the scheme thus performed worst in terms of correlation followed by variability, and these two components thus exert the dominant influence on the final KGE scores. This suggests that in order to get an improved KGE score of RH, the most important component score to improve is the correlation and, the next is the variability. The mean KGE value across

all other schemes for the 17 EREs is 0.20, with mean scores of 0.54, 0.18, and 0.43 for $|r-1|$, $|\beta-1|$, and $|\gamma-1|$, respectively. This indicates that, for all other scheme combinations, correlation and variability components predominantly influence the KGE values.



Figure 3d presents the values of KGE and its components ($r$, $\gamma$, and $\beta$; Eq. 1; Gupta et al., 2009; Kling et al., 2012) for all
17 events for the best performing scheme (Thompson-YSU; MP8_BL1) for WS. For the best scheme, the mean KGE score for
WS across 17 events is 0.29, while the mean scores for $|r-1|$, $|\beta-1|$ and $|\gamma-1|$ are 0.52, 0.28, 0.30, respectively. Among the
three KGE components, the scheme thus performed worst in terms of correlation and variability, and these two components
thus exert the dominant influence on the final KGE scores. This suggests that to get an improved KGE score of WS, the most
important component scores to improve are correlation and variability. The mean KGE value across all other schemes for the
17 EREs is 0.26, with mean scores of 0.57, 0.27, and 0.29 for $|r-1|$, $|\beta-1|$, and $|\gamma-1|$, respectively. These results indicate
that, for all other scheme combinations, the correlation and variability components have the most significant influence on the
KGE values.

### 4.9    Which BL and MP schemes were used in previous studies focusing on the Middle East?

**Table 5.** Studies simulating EREs in the Middle East using WRF.

| Study | MP Scheme | BL Scheme | Key Findings |
|---|---|---|---|
| Luong et al. (2020) | Morrison (MP10) | Mellor-Yamada-Janjic (MYJ; BL2) | Evaluated urbanization impacts on ERE over Jeddah; high-resolution models essential for urban storm simulation. |
| Francis et al. (2024) | Thompson aerosol-aware (MP28) | Quasi-Normal Scale Elimination (QNSE; BL4) | Enhanced performance in capturing precipitation patterns for events involving atmospheric rivers in the Middle East. |
| Deng et al. (2015) | Lin (MP2), Eta Ferrier (MP5) | Mellor-Yamada-Janjic (MYJ; BL2) | Demonstrated role of different MP schemes in capturing Jeddah flash-flood events. |
| Attada et al. (2020) | Thompson (MP8) | MYNN Level 3 (BL6) | Consistent performance in simulating rainfall events for AP EREs in arid regions. |
| Taraphdar et al. (2021) | Thompson (MP8) | Quasi-Normal Scale Elimination (QNSE; BL4) | Optimal pairing for precipitation simulation under 9-km resolution, balancing accuracy and efficiency in UAE simulations. |
| Abida et al. (2022) | WSM 3-class (MP3) | YSU (BL1) | Best performance in hyper-arid coastal regions, enhancing temperature, humidity, and wind accuracy at BNPP site. |
| Almazroui et al. (2018) | Eta Ferrier (MP5) | YSU (BL1) | Highlighted YSU's reliability for BL dynamics in extreme storm conditions (e.g., Jeddah 2009 event). |
| Patlakas et al. (2023) | Single-moment six-class (MP6) | YSU (BL1) | YSU's adoption in operational forecasting at the Saudi National Center for Meteorology for its robustness in arid climates. |





Although our findings are subject to uncertainty and must be interpreted with caution, as highlighted in the previous sub-sections, they provide a useful basis for evaluating schemes used in previous WRF studies in the region. Our review of these studies (Table 5) reveals varying choices of BL and MP schemes, with mixed alignment to the results of this study. Several studies, such as those by Abida et al. (2022), Almazroui et al. (2018), and Patlakas et al. (2023), used the YSU BL scheme (BL1), which our results confirm as the best-performing scheme for capturing the unique convective dynamics in arid climates. These studies highlighted YSU's robust vertical mixing capabilities and adaptability to desert environments. On the other hand, studies like Attada et al. (2020) and Taraphdar et al. (2021), which employed MYNN Level 3 (BL6) and QNSE (BL4), respectively, used local turbulence schemes that our findings show may be less suited for unstable, highly convective conditions typical in the region. Similarly, while MP schemes like Thompson (MP8) and Goddard (MP7), identified in our study as well-performing, were used in some cases (Taraphdar et al., 2021; Attada et al., 2020), other studies, such as Deng et al. (2015), relied on simpler MP schemes like Lin (MP2) and Eta Ferrier (MP5), which may lack the sophistication needed to capture mixed-phase processes in intense convective systems fully. Thus, while several studies used high-performing schemes, others could have benefitted from incorporating YSU and advanced MP schemes to enhance the accuracy of ERE simulations in this region. However, we would like to reiterate that our findings are subject to uncertainty, and these conclusions should therefore be interpreted with caution.

### 4.10  How generalizable are our findings?

We conducted the most comprehensive assessment of BL and MP schemes ever undertaken in the Saudi Arabia region. This study analyzed 17 EREs across the country and tested 36 different scheme combinations. In contrast, most prior studies focused on single events with a limited number of scheme combinations (Table 1). By conducting such an extensive evaluation, we were able to quantify the uncertainty in our results and highlight the challenges associated with these kinds of assessments. Additionally, our study represents a foundational reference for selecting the most suitable BL and MP schemes for simulating EREs in the region.

The findings of this study are particularly significant for regions with climate conditions similar to those in the AP. In desert regions with comparable features—such as low moisture availability, deep boundary layers, storms often driven by the passage of subtropical or polar jet streams, significant temperature variability, and unique land surface interactions—the combination of the YSU BL scheme with the Goddard and Thompson MP schemes is likely to perform effectively. This parameterization set could be a valuable test option for other arid or semi-arid regions with similar characteristics. However, further research may be necessary to fine-tune parameterization choices for accurate weather simulations in other areas.

### 5  Conclusion

This study evaluates the optimal combination of BL and MP parameterizations for simulating EREs at a convection-permitting resolution in the AP using the WRF-ARW model. 36 BL-MP combinations were evaluated over 17 ERE cases across the region. Our answers to the questions posed in the introduction are as follows:





1. Which BL scheme performs best in terms of rainfall?

   The YSU (BL1) scheme outperformed other schemes, achieving a temporal KGE of 0.43 and a spatial KGE of 0.29. This superior performance is attributed to non-local mixing, which enhances vertical transport and convective processes. This makes it particularly effective for simulating extreme rainfall in arid regions like Saudi Arabia. In contrast, local schemes such as MYNN and BouLac performed worse because they rely on small-scale turbulence, which limits the representation of deep convection.

2. Which MP scheme performs best in terms of rainfall?

   The Goddard (MP7) and Thompson (MP8) schemes performed the best, achieving a temporal KGE of 0.42, with spatial KGEs of 0.33 and 0.31, respectively. Their strong performance is attributed to their advanced mixed-phase and ice-phase MP. Thompson's double-moment structure enhances adaptability, while Goddard's optimized ice and graupel processes improve convective simulations. These results highlight the importance of advanced MP schemes for accurately modeling EREs in arid regions.

3. Which component of the Kling-Gupta Efficiency (KGE) affects the final score the most?

   Among the components of the KGE, correlation and variability significantly influenced KGE scores for precipitation. Enhancing these components could further improve the accuracy of ERE simulations.

4. How statistically significant are the differences in performance between scheme combinations in terms of rainfall?

   Pairwise statistical tests revealed that the YSU (BL1) and Thompson (MP8) combination significantly outperformed 21 other scheme combinations, while the poorest-performing scheme, Morrison-MYNN (MP10_BL6), was statistically inferior to 28 other combinations. This confirms that the selection of schemes plays a critical role in model accuracy.

5. How consistent are the temporal and spatial performance assessments for rainfall?

   The assessment reveals that the Goddard (MP7) and Thompson (MP8) MP schemes, combined with the YSU (BL1) BL scheme, performed best in both temporal and spatial KGE evaluations. The higher mean temporal KGE (0.37) compared to the spatial KGE (0.26) indicates that the model captures rainfall variability over time more effectively than its spatial distribution. Although spatial KGE values were lower, the order of scheme performance remained consistent.

6. How consistent is the performance ranking among different variables?

   We obtained weak correlations between rainfall performance and other variables, indicating poor consistency. This suggest that different physical processes govern the simulations of different variables. While MP and BL schemes influence precipitation, other components, such as land surface and radiation schemes, may affect temperature and wind. This underscores the complexity of model parameterization and the need for further research.

7. What do the spatial patterns in simulated and observed rainfall look like for the events?





The spatial patterns of simulated and observed rainfall captured well but exhibited occasional overestimations and underestimations. These discrepancies are likely due to boundary condition limitations (ERA5 forcing) and satellite data uncertainties in the IMERG reference dataset.

8. How well does the model perform in terms of the other variables?

The Thompson-YSU (MP8_BL1) scheme provided the best overall results for additional variables, including 2-meter
temperature, 2-meter relative humidity, and wind speed. This suggests that it is a robust scheme choice for broader meteorological applications in desert environments.

9. Which BL and MP schemes were used in previous studies focusing on the Middle East?

Many previous studies in the Middle East have employed BL and MP schemes that align with our findings, confirming the robustness of the YSU scheme for BL dynamics. However, some past studies using simpler MP schemes, such as
Lin (MP2) and Eta Ferrier (MP5), may have benefited from adopting more advanced schemes like Thompson (MP8) for improved simulation accuracy.

10. How generalizable are our findings?

With an extensive evaluation of 36 scheme combinations across 17 EREs, this study serves as a foundational reference for selecting BL-MP schemes in desert environments. The results mainly apply to regions with similar climatic con-
440 ditions, characterized by deep BLs, intense surface heating, and moisture-limited convection, significantly influencing precipitation processes. Future studies incorporating radar data would refine these insights and enhance model accuracy.

By identifying the optimal BL and MP combination from 36 tested configurations across 17 EREs, the study establishes a strong foundation for improving the accuracy of ERE projections. As the most comprehensive evaluation of BL and MP schemes in Saudi Arabia to date, this research provides valuable insights into how parameterization choices affect ERE simu-
445 lations. In a region that remains underexplored despite frequent flash floods and significant casualties, these findings serve as a key reference for future modeling efforts. The results guide researchers and forecasters in selecting effective parameterization schemes, ultimately contributing to more reliable forecasting and enhanced disaster preparedness in arid environments.

*Code availability.* The code used to generate the results of this study is available from the corresponding author upon request.

*Data availability.* The IMERG precipitation data are available via NASA GES DISC(https://disc.gsfc.nasa.gov/datasets/GPM_3IMERGHH_
07/summary?keywords=GPM_IMERG), while the ERA5 data is available at the Copernicus Climate Data Store (CDS; https://cds.climate.
copernicus.eu). The radiosonde data available on the University of Wyoming's upper-air sounding archive (https://weather.uwyo.edu/upperair/
sounding.html) while surface meteorological data is accessible via the Iowa State University Environmental Mesonet website (https://
mesonet.agron.iastate.edu/request/download.phtml?network=SA__ASOS).



*Author contributions.* RKS: modeling, analysis, visualization, and writing. HB: initial idea, conceptualization, writing, oversight of analysis, and project administration. All coauthors contributed to writing, revising, and refining the manuscript.

*Competing interests.* The authors declare that they have no conflict of interest.

*Acknowledgements.* For part of our analysis, we used resources from the Shaheen supercomputer at King Abdullah University of Science and Technology (KAUST) in Thuwal, Saudi Arabia. The authors acknowledge the European Center for Medium-Range Weather Forecasts (ECMWF), National Aeronautical and Space Administration (NASA), University of Wyoming, and Iowa state university for the ERA5, GPM, radiosonde, and METAR datasets, respectively.



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
