# Peer review of "Evaluation of Microphysics and Boundary Layer Schemes for Simulating Extreme Rainfall Events over Saudi Arabia using WRF-ARW v4.4"

_EGUsphere, 2025_

## Referee Comment (RC1)

[referee-annotated manuscript omitted]

---

## Referee Comment (RC3)

**Referee Comments for the article titled "Evaluating Microphysics and Boundary Layer Schemes in WRF: Assessment of 36 Scheme Combinations for 17 Major Storms in Saudi Arabia"**

The present article provides a comprehensive literature review in the field of urban meteorology, with a particular focus on the challenges and developments over Saudi-Arabia and middle-east.

The review work by Sahu Rajesh et al. (2025) is a well-prepared and valuable manuscript that provides a detailed review of evaluating microphysics and boundary layer schemes in WRF over Saudi Arabia. The authors have done an excellent job in addressing extreme rainfall events and its modelling over data sparse region like Saudi Arabia. I think that the manuscript presents a well-structured and comprehensive review with significant improvements in clarity, depth, and organization. Though the manuscript is well structured and presented, I have a very few minor comments related with the manuscript, which I feel that the authors should incorporate. The manuscript has the potential to get acceptance, only after addressing the comments below:

**Major/Minor Comments**

1. Consider making the title slightly more concise and catchier. For example:

"Evaluation of Microphysics and Boundary Layer Schemes for Simulating Extreme Rainfall events over Saudi Arabia using WRF".

2. In Abstract kindly rephrase the line: "Kling-Gupta Efficiency (KGE) incorporates correlation, variability, and overall bias."

to

 "The Kling-Gupta Efficiency (KGE) metric, which incorporates correlation, variability, and bias, was used for performance evaluation." Provide necessary (original WMO) citations for the metrics used.

3. In section 1 where you structure the ten key questions, consider using letters (a, b, c...) for questions to avoid confusion with numbered sections.

4. Ensure all acronyms (e.g., MP, BL, KGE, IMERG) are defined on their first use in both the abstract and main text.

5. Add more region-specific references: While many global references are cited, consider including more recent or specific studies on EREs or WRF performance over Saudi Arabia or the Middle East (e.g., 2022–2024 publications if available).

6. On page 2, after line number 25 rephrase the line "These events are often linked to the intrusion of intensified subtropical jet stream…" to:

"These events are frequently associated with intrusions of an intensified subtropical jet stream…"

7. Make the figure captions of Figure 2, 3 and 4 more self-explanatory by specifying metrics, datasets, and periods used.

8. In the abstract section after line number 10, "The Thompson-YSU combination yielded the highest mean KGE…"

Rephrase to "Among all 36 combinations, the Thompson-YSU pairing consistently produced the highest mean KGE across the 17 storm events."

9. Ensure consistent use of terms like "EREs," "events," "storms" throughout the paper. Stick with one preferred term unless differentiation is needed.

10. After line number 285, "...models often struggle to replicate the spatial distribution of events precisely."

Suggestion is to rephrase: "This is expected, as localized convective systems common in the region present challenges for accurately resolving spatial rainfall patterns in mesoscale models."

11. After line number 290: "...the Goddard (MP7) and Thompson (MP8) MP schemes, particularly when paired with the YSU (BL1)…... emerged as superior."

Rephrased to "...the Goddard (MP7) and Thompson (MP8) schemes, when combined with YSU (BL1), consistently ranked highest across both temporal and spatial KGE assessments."

12. When discussing major findings (e.g., Thompson–YSU being best), consider referencing the figure or table that supports this claim.

13. The paper can be redrafted to explain the section 4.7 in the beginning i.e. before section 4.1. This is so that readers gets a visual demonstration of the rainfall event in the domain of the study.

14. In the conclusion section of the study bring out the motivation/conclusion of the study that this is a kind of a verification study for hydrometeorology.

15. The authors can also verify the 850hPa wind and near surface temperature and provide plots in supplementary section.

---

## Author Comment (AC1)

**Pointwise replies to reviewer's comments on the manuscript "Evaluating Microphysics and Boundary Layer Schemes in WRF: Assessment of 36 Scheme Combinations for 17 Major Storms in Saudi Arabia" (egusphere-2025-912)**

**Response to the comments of Reviewer 1**

We thank the reviewers for providing such positive and detailed suggestions regarding the manuscript. We have improved our work according to their concerns.

**Comment 1:** Please use standard terminology (e.g., "Planetary Boundary Layer – PBL" or "convection-permitting").

**Response:** Thank you for the constructive suggestion. We have revised the manuscript to consistently use standard terminology, such as "Planetary Boundary Layer (PBL)" and "convection-permitting resolution." All acronyms are now defined at their first occurrence to ensure clarity for the reader.

**Comment 2:** The way of phrasing the aims of the study (i.e., using research questions) is not appropriate for a scientific article. In addition, some of these questions are not so relevant and could be removed (e.g., #10." How generalizable are our findings?"). The same applies to section or sub-section headings, and the Conclusions, where I would avoid using questions.

**Response:** Thank you for this thoughtful comment. While we recognize the traditional preference for clearly stated aims over question-based formats, using research questions is common in interdisciplinary and applied studies. After careful consideration and discussion amongst the co-authors, we believe our question-driven structure effectively conveys the study's objectives and findings, particularly given the breadth and complexity of our analysis. However, we removed the Question #10 ("How generalizable are our findings?") as per your suggestion. We remain open to reformatting the structure if the editor prefers.

**Comment 3:** More information should be provided on why the selected processes (PBL and cloud microphysics) were investigated. How do these parameterizations influence the simulation of precipitation? For example, elaborate on what a single and a double moment scheme is. Are there other processes relevant (e.g., convection in the coarser-resolution domain)?

**Response:** We appreciate the reviewer's comment. In response, we have expanded the introduction to clarify the rationale for focusing on the PBL and cloud microphysics (MP) schemes. Specifically, we added the sentence: "Two key parameterization schemes that strongly influence ERE simulations include the Planetary Boundary Layer (PBL) and cloud microphysics

(MP) schemes." We then included two dedicated paragraphs: one describing the role of PBL schemes in modulating turbulent mixing, boundary-layer growth, and the initiation of convection; the other describing how MP schemes govern hydrometeor development, cloud phase transitions, and precipitation intensity. The MP section also explains single- and double-moment schemes, including their implications for predicting hydrometeor number concentrations and mass. Each paragraph contains citations to relevant literature.

We acknowledge that other processes — such as cumulus parameterizations in the coarser domain, radiation schemes, and land surface interactions — also affect precipitation simulation. However, we limited the present assessment to PBL and MP schemes because they exert the most direct control on convective processes and cloud microphysics at convection-permitting scales, as well as to limit computational and storage demands. We have clarified this scope limitation in the conclusion, adding: "To further advance ERE simulation fidelity, future work should extend beyond PBL and MP schemes to systematically evaluate the impact of land surface schemes, radiation parameterizations, and data assimilation techniques." (Lines 40-49, 51-58)

**Comment 4:** The selection and definition of extreme cases is problematic, since for many events there is unknown information on the observed precipitation amounts (Table 2). I recommend including an additional column which will show the IMERG nearest grid-cell precipitation.

**Response:** Thank you for your valuable comment. We have added a new column with IMERG rainfall amounts for the events.

**Comment 5:** It is also unclear if any events lasted more than one day, and if yes, how were these events treated in the analysis? Is one day of spin-up time enough for these runs?

**Response:** Thank you for raising this critical point. In all cases, the large majority of rainfall occurred within one day. All model simulations were conducted for 84 hours, including a 48-hour spin-up period to ensure model stability and reduce initialization biases. The analysis was focused on a 24-hour window corresponding to the peak rainfall period of each extreme event (Table 2). Our study specifically targets short-duration, event-based simulations of extreme rainfall. In such cases, the primary drivers are typically large-scale atmospheric instabilities and moisture advection rather than slower processes like land–surface interactions. Consequently, a 48-hour spin-up period is sufficient to allow the model to dynamically and thermodynamically adjust to the initial and boundary conditions. This clarification has been added to the revised Data and Methods section (Line 139-145).

**Comment 6:** For extracting the overall statistics, all events were weighted equally. However, in the interpretation of results, it would have been useful to differentiate, for example, between the most and the less extreme events, or between events affecting different parts of the Arabian Peninsula.

**Response:** Thank you for this insightful suggestion. We fully agree that distinguishing between more and less extreme events, as well as regional variability, can offer valuable insights. However, in our case, we found no systematic dependence of KGE values on the rainfall intensity of the events (see the figure below). There is no discernible trend or correlation between IMERG rainfall amounts and the corresponding KGE values, suggesting that model performance does not scale with event intensity. Based on this finding, we consider that giving equal weight to all events is a reasonable and justified approach in our statistical summaries. Additionally, stratifying the results further would make an already complicated analysis even more, hindering interpretation and presentation.

[Figure]

**Comment 7:** More information on the interpretation of KGE should be provided in Section 3.5. Some references to other studies that use KGE in a spatial context could also be added. Moreover, I strongly recommend using additional evaluation metrics and not relying only on KGE for your conclusions.

**Response:** Thank you for the suggestion. We have clarified in Section 3.5 that KGE is an aggregate metric incorporating correlation, bias ratio, and variability ratio. To maintain clarity and avoid overwhelming the reader, we chose not to include additional performance metrics. However, we have added the following sentence to support our approach and provide references for spatial applications of KGE:

"The KGE is an aggregate performance metric that integrates correlation, bias ratio, and variability ratio into a single score, providing a holistic assessment of model performance. While additional metrics could be computed, including too many would risk overwhelming the interpretation. Several studies have successfully used KGE for spatial performance assessment of hydrometeorological models (e.g., Gupta et al., 2009; Patil and Stieglitz, 2014; Beck et al., 2019; Nguyen et al., 2022; Tudaji et al., 2025), supporting its application in our spatial analysis." (Line 148-151)

**Comment 8:** Extensive parts of Section 4 are not results (e.g., L148-161, L217-222, L275-280, L326-332). Please move this and other non-results material to the introduction, data or discussion sections, if relevant.

**Response:** Thank you for your observation. We would like to clarify that Section 4 is the Results and Discussion section. Accordingly, besides presenting the results, we provided interpretation, compared our results to other studies, and answered the questions posed in each subsection. Nevertheless, we have carefully reviewed Section 4 and agree that Lines 148–161 contain background context and methodological details. These have now been relocated to the introduction (Line 43-49). Regarding Lines 217–222, 275–280, and 326–332, we respectfully retain these in their current positions, as these paragraphs serve as essential contextual discussion that supports the interpretation of findings.

**Comment 9:** The approach described in lines 251-257 should be presented in more detail in the Methods section.

**Response:** Thank you for pointing this out. We agreed and have expanded the section as follows:

"Additionally, to determine whether the performance is significantly different between scheme combinations, we calculated $\Delta$KGE scores by subtracting the mean KGE across events from the KGE values, thereby eliminating systematic differences in scores among events. We then tested whether the distributions of $\Delta$KGE values for different scheme combinations are statistically similar or different using pairwise independent t-tests." (Line 166-169)

**Comment 10:** Figures 6 and 7 should be merged to facilitate the comparison between observations and simulated rainfall. Please be consistent in the date format (panel titles).

**Response:** Thank you for the suggestion. We strongly considered merging Figures 6 and 7 to facilitate direct comparison; however, doing so would result in 34 panels, which we believe would compromise clarity and interpretability. Instead, we have ensured that the date format is now consistent across all panel titles to improve the consistency and facilitate easier comparison between the figures.

**Comment 11:** Sections 4.9 and 4.10 are definitely not results material. Please move to other more relevant section(s).

**Response:** Thank you for your feedback. Our paper includes a merged Results and Discussion section for improved readability. Sections 4.9 and 4.10 were originally intended to revisit the research questions and synthesize key insights. However, we acknowledge these were more interpretative in nature. We have retained Section 4.9, as it revisits the research questions and synthesizes key findings in a way that supports the overall coherence of the paper. Section 4.10 has been removed as suggested.

---

## Author Comment (AC3)

**Pointwise replies to reviewer's comments on the manuscript "Evaluating Microphysics and Boundary Layer Schemes in WRF: Assessment of 36 Scheme Combinations for 17 Major Storms in Saudi Arabia" (egusphere-2025-912)**

**Response to the comments of Reviewer 2**

We are very grateful for the valuable suggestions from the reviewer. The changes addressed by the reviewer have been carefully incorporated into the manuscript in the following manner:

**Comment 1:** Please make sure that you clearly distinguish between "schemes" and combinations as this is very confusing for the reader.

**Response:** Thank you for the feedback. We went through the paper once again and ensured we used the right terminology throughout the manuscript: schemes for individual physical parameterization schemes (e.g., specific PBL or MP schemes) and combinations for scheme combinations (e.g., a specific pairing of PBL and MP schemes).

**Comment 2:** Please make sure that you do not repeat the explanations of the MP and BL option combinations. Explaining them once and then use "MP8_BL1" throughout the manuscript should be sufficient.

**Response:** Thank you for your suggestion. We have revised the manuscript to remove repeated explanations of the MP and PBL option combinations.

**Comment 3:** Figure captions in the continuous test should start with "Fig." instead of "Figure". "Figure" is only used at the beginning of a sentence.

**Response:** Thank you for the clarification. We have revised all figure references accordingly. We now use Fig. or Figs. consistently.

**Comment 4:** What is your final recommendation regarding the combination of PBL and MP? This should be mentioned as this is an important outcome of your study.

**Response:** Thank you for highlighting this important point. We agree that the final recommendation regarding the optimal combination of PBL and MP schemes is a key outcome of the study and should be clearly stated, and we have clearly emphasized this recommendation in the abstract, conclusion, and relevant discussion sections of the manuscript (Line 11-12, 380-391).

**Comment 5:** Manuscript title: Please include the WRF version you are using as different WRF versions can lead to different results.

**Response:** We appreciate the suggestion, and agree that different versions can lead to different results. We have added the version (WRF-ARW v4.4) in the manuscript title.

**Comment 6:** Line 5: I think "convection-permitting" is more widely used than "convective-permitting".

**Response:** Thank you for the suggestion. We agree and have revised "convective-permitting" to "convection-permitting" throughout the manuscript to align with widely accepted terminology and ensure consistency (Table 1, Line 6, and 67).

**Comment 7:** Line 11: Where are the "21" combinations are coming from? The abstract suggest you performed 36 combinations.

**Response:** Thank you for your observation. The sentence refers to the results presented in Section 4.4, where statistical testing demonstrated that the Thompson–YSU combination performed significantly better than 21 out of the 36 scheme combinations evaluated in the study. We have clarified this point in the manuscript to avoid any ambiguity.

**Comment 8:** Line 20: Are there more recent publications which cover the aspect of climate change?

**Response:** Thank you for the comment. We acknowledge that more recent studies addressing the impacts of climate change on extreme rainfall events (EREs) are available. Accordingly, we have updated the references in Line 20 to include recent and relevant publications, thereby strengthening the contextual foundation of our study (Line 23).

Muller, C., & Takayabu, Y. (2020). Response of precipitation extremes to warming: what have we learned from theory and idealized cloud-resolving simulations, and what remains to be learned?. Environmental Research Letters, 15(3), 035001.

Fowler, H. J., Lenderink, G., Prein, A. F., Westra, S., Allan, R. P., Ban, N., ... & Zhang, X. (2021). Anthropogenic intensification of short-duration rainfall extremes. Nature Reviews Earth & Environment, 2(2), 107-122.

Neelin, J. D., Martinez-Villalobos, C., Stechmann, S. N., Ahmed, F., Chen, G., Norris, J. M., ... & Lenderink, G. (2022). Precipitation extremes and water vapor: Relationships in current climate and implications for climate change. Current Climate Change Reports, 8(1), 17-33.

**Comment 9:** Line 33: "can feed early warning systems"

**Response:** Thank you for the suggestion. We have revised the sentence and believe to enhance clarity and improve readability (Line 31-33).

**Comment 10:** Line 36: "inform" seems not an appropriate word here.

**Response:** Thank you for the feedback. We have revised the sentence (Line 33).

**Comment 11:** Line 37: The acronyms "AP" and "WRF" are not explained. Please ensure that all acronyms are explained before they are used in the manuscript. Also add the reference for the WRF model here.

**Response:** Thank you for the helpful comment. We have defined the acronyms "AP" (Arabian Peninsula) and "WRF-ARW" (Advanced Research version of the Weather Research and Forecasting) at their first occurrence in the manuscript. In addition, we have included the appropriate reference for the WRF model in this section to ensure proper attribution and context (Line 34-35).

**Comment 12:** Line 37: "Numerical Weather Prediction (NWP) model"…

**Response:** Thank you for the comment. We have revised the sentence to introduce WRF as a "Numerical Weather Prediction (NWP) model" (Line 35).

**Comment 13:** Line 41: The microphysics also have an impact on radiation.

**Response:** Thank you for the suggestion. In response, we have revised the sentence in the manuscript to: "The MP scheme controls cloud formation, precipitation processes, and interactions between different water phases. It also influences radiative transfer by affecting cloud optical properties such as droplet size distribution, phase, and concentration." (Lines 51-53)

**Comment 14:** Line 53: "to evaluate the best combination"….

**Response:** Thank you for the suggestion. As recommended, we have rephrased the sentence from "to establish the best combination" to "to determine the best combination" as we felt "determine" was more fitting (Line 67).

**Comment 15:** Line 56: "using WRF in a two-way nested"… What was your motivation to apply a two-way nesting approach?

**Response:** Thank you for the comment. We used a two-way nesting approach to allow feedback between the high-resolution inner domain and the coarser parent domain. This is essential for capturing small-scale processes like convection, PBL turbulence, and orographic effects, which can influence larger-scale circulation. The dynamic interaction improves physical consistency and

is crucial for realistically simulating mesoscale convective systems (MCS) and associated precipitation.

**Comment 16:** Line 66: I do not think that number 9 is a key question of your study.

**Response:** We appreciate the comment. However, after careful consideration, we have decided to retain question number 9, as we believe it complements the broader context of our study. However, we have clarified its relevance to the core objectives within the discussion to ensure focus and coherence.

**Comment 17:** Line 69: "spans from 16°N to 33°N and 34°E to 56°E"

**Response:** Thank you for the suggestion. We have revised the statement (Line 83).

**Comment 18:** Starting line 73: The readers may be not familiar with all the different regions of the Arabian Peninsula and Saudi Arabia. I think it would be good to add at least some of the regions and major cities you mentioned to Fig. 1.

**Response:** We have updated Figure 1 to include key locations such as Riyadh, Jeddah, Hafr Al-Batin, Tabuk, Mecca, Medina, Najran, Jizan and Abha, supporting easier interpretation of the results (Line 92-93).

**Comment 19:** Table 1: The study of Schwitalla et al. (2020) is the only experiment in your table with a convection-permitting model resolution. This should be mentioned in the table itself (maybe as a separate column) and/or in the text. Please consider to add the number of model layers of the different studies as this can help the reader to further interpret your findings.

**Response:** Thank you for your valuable suggestions. We have updated Table 1 to highlight that the study by Schwitalla et al. (2020) is the only one among the listed references that employed a convection-permitting model resolution. Additionally, we have added a new column titled "Model layer/ Vertical levels used" to indicate the number of model layers used in each study, where such information was available. We have also added a discussion on this particular aspect in the main text (Line 205-213).

**Comment 20:** Line 94: Did you use ERA5 pressure or model level data for the initialization of the model? This has influence on the accuracy of the initial conditions as the number of model levels in ERA5 is about 100 up to 30 km while the number of pressure levels is 32 up to approximately. 30 km altitude. If you use pressure level data, please explain your decision.

**Response:** We appreciate the insightful comment. We used ERA5 pressure-level data (37 levels) for initializing and forcing the WRF model. While ERA5 model-level data are available and offer higher vertical resolution (137 levels), we opted for pressure-level data due to computational and practical constraints.

Our experiment involved 612 high-resolution (3 km) convection-permitting simulations – derived from 36 physics scheme combinations across 17 EREs. Incorporating model-level data in this context would have significantly increased data volume, preprocessing time, and simulation runtime, making the overall workflow infeasible given our available resources. Moreover, our goal was to assess inter-scheme sensitivity in precipitation simulation, for which pressure-level data have been used widely and successfully.

**Comment 21:** Line 101: Radiosonde stations are plotted in Fig. 1 but were never used or described explicitly throughout the manuscript. Or are they included in the analysis shown in sec. 4.6? If yes, I think it is dangerous to combine them together as you would combine prognostic (3D) variables with diagnostic (2D) variables. Did you consider differences between the station altitude and the model orography in case stations are located in the mountains? If this difference is large, this can alter your results for T, RH, and WS.

**Response:** Thank you for your detailed observation. We acknowledge the potential confusion caused by the inclusion of radiosonde station locations in Figure 1. To clarify, we did not use any radiosonde data at any point in the study—including in Section 4.6. The radiosonde locations were initially included in Figure 1 for geographical context, but since they are not part of the analysis, we agree that their inclusion may be misleading. In response to your suggestion, we have removed the "radiosonde locations" from Figure 1, and the updated figure now shows only the METAR station locations.

The surface observations used in Section 4.6 are exclusively from METAR stations, obtained via the Iowa Environmental Mesonet (https://mesonet.agron.iastate.edu/request/download.phtml?network=SA__ASOS). These datasets provide 2D diagnostic variables such as temperature (2 m), relative humidity (2 m), and wind speed (10 m), which are used to validate the WRF model outputs at corresponding levels. We also agree with your concern that combining 3D radiosonde data with 2D surface diagnostics, particularly in mountainous regions, can lead to significant errors if altitude differences are not accounted for. Since we use only METAR surface data, such altitude mismatches between station height and model terrain are not relevant in our case.

**Comment 22:** Line 108: WRF version 4.4.0?

**Response:** Thank you for pointing this out. We confirm that all simulations were conducted using WRF version 4.4, and the manuscript has been updated to mention this in the abstract and the Data and Methods section (Line 119).

**Comment 23:** Line 110: Please add the corresponding number of grid cells and the model top pressure here and to Table 3.

**Response:** Thank you for the suggestion. The model top pressure used in our simulations was 3000 Pa (30 hPa). The number of horizontal grid cells was 493 × 418 for the parent domain (D01) and 1012 × 889 for the nested domain (D02). We have updated both the text at line 110 and Table 3 to include this information (Line 120-122).

**Comment 24:** Line 120: The Kessler scheme is an extremely old scheme. Please consider whether it is necessary to apply this in your study on a convection-permitting resolution. It is preferably used in idealized cloud modeling studies.

**Response:** Thank you for the insightful comment. We acknowledge that the Kessler scheme is a relatively simpler microphysics parameterization, primarily designed for warm-rain processes and often used in idealized cloud modeling studies. However, in our study, we included it as a baseline reference to compare the performance of more advanced MP schemes. This allows us to better highlight the improvements offered by more physically comprehensive schemes under convection-permitting resolutions.

**Comment 25:** Line 130: the simulations were run for 84 hours with a 48 hour spin-up followed by a 24h forecast. Why did you run the model for 84 hours? Did you reinitialize the atmosphere after your 48h spin-up?

**Response:** Thank you for your question. The model was run continuously for 84 hours without reinitialization. The first 48 hours were treated as a spin-up period, and the following 24-hour window (hours 49–72) was used for evaluation and comparison with observations.

The additional 12 hours (hours 73–84) were included to allow flexibility in capturing the full evolution of EREs, particularly in cases where the peak precipitation may extend slightly beyond the 72-hour mark. This precaution ensured that the forecast window did not truncate significant rainfall signals near the end of the evaluation period. However, only the 24-hour forecast period (49–72 h) was used for performance assessment in this study.

**Comment 26:** Figure 1: Please use a different color table as there is too much blue color in the figure. Depending on the printer, it can be difficult to see the red markers on a blue background. Also, add the METAR stations here as this is important for the interpretation of your results.

**Response:** Thank you for your helpful suggestion. We have revised Figure 1 and changed the colors to enhance clarity and accessibility. Additionally, we have included the locations of the METAR stations used in our analysis.

**Comment 27:** Table 2: It would be great if you could add the rainfall amount for all cases. Otherwise it is difficult for the reader to judge the level of extreme of the particular event. It really matters if you have 200 mm within a day of more than 40 mm in 30 min.

**Response:** Thank you for the valuable suggestion. We agree that including rainfall amounts is important for evaluating the severity of each event, as also suggested by Reviewer #1. Accordingly, we have added a new column to Table 2 showing the IMERG maximum rainfall for each case.

**Comment 28:** Table 3: Does the Arakawa-C grid play a role in your study? As far as I know it cannot be changed anyway. Also, the reference for the NOAH LSM is missing here. Regarding the CU scheme: I think you mean that for D02 no CU scheme is used.

**Response:** You are correct that the Arakawa-C grid is a fixed configuration in WRF and cannot be modified by the user. Accordingly, we have removed this entry from Table 3, as it does not provide value to the reader. We have also added the appropriate reference for the NOAH land surface model (Chen and Dudhia, 2001) to both the table and the references section. Regarding the cumulus parameterization (CU) scheme, we confirm that it was applied only in the parent domain (D01), while no CU scheme was used in the nested domain (D02). This clarification has been added to Table 3 to prevent any confusion.

**Comment 29:** Regarding the model setup: Did you use the default data sets for soil texture and land cover? This should be mentioned either in Table 3 or in the text.

**Response:** Yes, we used the default WRF datasets for both soil texture and land cover. Specifically, the United States Geological Survey (USGS) 21-category land use dataset and the default soil texture data provided with the WRF Preprocessing System (WPS) were employed. We have updated the manuscript to include this information in both the model setup description and Table 3.

**Comment 30:** Line 140: Please add a short sentence about what a perfect KGE would be. I guess 1 is ideal. What does a KGE of 0.5 tell us (most of the values are below this threshold)? Did you interpolate the simulated precipitation from the WRF model to the IMERG grid? If it was done the other way round, I doubt that this is meaningful. Which interpolation method did you use (bilinear, conservative, nearest neighbor, etc.)? Please also mention the tool or software package used for interpolation.

**Response:** A perfect Kling-Gupta Efficiency (KGE) score is 1.0, indicating perfect agreement between simulated and observed data. A KGE of 0.5 suggests good agreement, given that a hypothetical baseline simulation predicting only the mean would achieve a KGE of -0.41 (Knoben et al., 2019), making our results quite reasonable, situated between this baseline and an (unattainable) perfect score (KGE of 1). To enable a grid-point-to-grid-point comparison with IMERG observations, we resampled the WRF-simulated precipitation data to the IMERG grid using averaging. This interpolation was performed using the xarray package in Python (Hoyer and Hamman, 2017). We have added these clarifications to the manuscript to enhance the transparency and reproducibility of our methodology (Line 156-158, 163-165).

Knoben, W. J., Freer, J. E., & Woods, R. A. (2019). Inherent benchmark or not? Comparing Nash–Sutcliffe and Kling–Gupta efficiency scores. Hydrology and Earth System Sciences, 23(10), 4323-4331.

Hoyer, S., & Hamman, J. (2017). xarray: ND labeled arrays and datasets in Python. Journal of Open Research Software, 5(1), 10-10.

**Comment 31:** Line 142: Isn't γ the ratio of the variances? "Coefficient of variation" sound a bit inappropriate.

**Response**: Thank you for the comment. γ is not the ratio of variances, but rather the ratio of the coefficients of variation (CV), which accounts for both standard deviation and mean. We agree that the term "coefficient of variation" alone may be unclear in this context, and we will revise the text to explicitly state that γ represents the ratio of the coefficients of variation of the simulated and observed data to improve clarity (Line 156-157).

**Comment 32:** Line 145: What was your motivation to use the nearest model grid cell next to the surface observations instead of, e.g., a distance-weighted 3x3 average? I think you cannot expect that the model can simulate your observation 1:1.

**Response:** We used the nearest model grid cell to each surface observation station to preserve a direct spatial correspondence and retain the raw, localized model signal. While we acknowledge that a distance-weighted 3×3 average could smooth out local variability and possibly offer a more representative comparison in complex terrains or heterogeneous conditions, our primary goal was to evaluate the model's point-level performance without introducing additional spatial averaging.

Moreover, we tested both approaches (nearest-neighbor and 3×3 distance-weighted average) and found that the latter did not result in a significant improvement in our evaluation metrics. Therefore, for consistency and clarity, we proceeded with the nearest grid point approach.

**Comment 33:** Lines 148-161: Please consider to integrate both paragraphs (or parts of them) to the introduction. I think it better fits there rather than in the discussion section.

**Response:** Thank you for your suggestion. We agree that integrating the discussion of PBL and MP schemes (Lines 148–161) into the introduction would offer readers earlier context regarding their significance in simulating EREs. We have revised the manuscript accordingly (Line 43-49).

**Comment 34:** Line 169-170: you mention that both MYNN schemes and the BouLac scheme rely on the representation of gradients. As you only have coarse 53 model levels, could the explain the detrimental performance of the MYNN and BouLac simulations?

**Response:** Thank you for this observation. As noted, both MYNN and BouLac, which are TKE-based and largely rely on local vertical gradients to accurately represent turbulent mixing. In our

setup, we used 53 vertical levels, which is a commonly adopted configuration. While this resolution provides a reasonable balance between detail and computational cost, it may not capture fine-scale vertical structures to the same extent as higher-resolution configurations.

We have already addressed this aspect in the manuscript (Lines 209-210), where we contrast our findings with Schwitalla et al. (2020), who used 100 vertical levels and reported better performance for the BL5 scheme. We note that differences in vertical resolution, along with ERE characteristics and surface conditions, likely contributed to the contrasting performance of BL5. This context has now been further clarified in the revised text to ensure alignment with the discussion on PBL scheme performance.

**Comment 35:** Line 184: What is a "Stratified" BL?

**Response:** Thank you for pointing this out. The term "stratified" BL refers to a BL with stable stratification—typically observed at night or over surfaces experiencing strong cooling—where vertical mixing is limited due to temperature inversions. To improve clarity and avoid ambiguity, we have revised the wording in the manuscript to "stably stratified BL" which more accurately describes the underlying physical process.

**Comment 36:** Line 185: This is in contrast to the study of Schwitalla et al. (2020) which you mentioned in Table 1.

**Response**: Thank you for the excellent point. Schwitalla et al. (2020) examined a single summertime convective event on 14 July 2015 over the Arabian Peninsula using a physics ensemble analysis. In contrast, our study covers 17 EREs across Saudi Arabia. This larger sample size enhances the statistical significance of our findings and tests scheme robustness under varied synoptic conditions.

Therefore, our identification of the YSU PBL and Thompson MP combination as optimal is based on consistent performance across multiple independent events, making it more generalizable than conclusions drawn from a single-case study.

We have now clarified this distinction in the revised manuscript (Line 205-214)

**Comment 37:** Line 201-204: I think this extensive explanation is not necessary here. You may introduce the abbreviations already in Table 3. This allows for saving space.

**Response:** Thank you for the suggestion. We agree that the extended explanation in the main text is redundant, given that the abbreviations are already clearly defined in Table 3. We have removed the full forms from this section and now refer only to the abbreviations (Line 218).

**Comment 38:** Line 205: "MP" is used for a microphysics scheme. Please write "cloud microphysics" instead.

**Response:** Thank you for the suggestion. To enhance clarity and avoid ambiguity, especially for readers who may not be familiar with the abbreviation, we have revised the wording to replace "MP" with "cloud microphysics (Line 219-220)."

**Comment 39:** Line 217: See my comment to line 201-204.

**Response:** Thank you once again for your helpful comment. As noted in our response to the comment about Lines 201–204 of the manuscript, the full forms of the abbreviations are already provided in Table 3. We have removed the repeated explanations from this section as well (Line 231).

**Comment 40:** Line 218: Which advanced microphysical processes?

**Response:** By "advanced microphysical processes," we refer to the inclusion of detailed representations such as including graupel and hail processes, multiple ice-phase species, prognostic treatment of various hydrometeors, and more complex interactions between cloud and rainfall particles (Line 232-233).

**Comment 41:** Line 221: Which sensitivities?

**Response:** We have rewritten this part to enhance the clarity and readability. The revised text does not include "sensitivities."(Line 235-237).

**Comment 42:** Line 230: I guess you mean "combination" instead of "scheme".

**Response:** Thank you for pointing this out. Corrected (Line 245).

**Comment 43:** Lines 231-232: Do the numbers show the absolute value of the mean of "r-1"? This is a bit confusing to the reader. Regarding the KGE: Did you account for a potential "double-penalty" of your model? In case the extreme precipitation is shifted by one grid cell in the model, the KGE may deteriorate.

**Response:** Thank you for the comment. This is not entirely correct. As mentioned in the text, the numbers represent the mean of the absolute values of r-1. This calculation was necessary to compare the values of r, β, and γ. Regarding the double-penalty effect, the KGE is sensitive to it, through the r. However, the β and γ components of the KGE are not sensitive to this.

**Comment 44:** Figure 3: What do you mean with "long-term bias" in the caption of Figure 3? In the second line of the figure caption, it should be "combination" instead of "scheme".

**Response:** In the caption of Figure 3, the term "long-term bias" refers to the bias component (β) of the KGE, which quantifies the ratio of mean simulated to mean observed precipitation and other meteorological parameters over the evaluation period. To enhance clarity, we have revised the

caption to use the term "bias" instead of "long-term bias." Additionally, we have replaced "scheme" with "combination."

**Comment 45:** Line 247: "(See later Fig. 5)".

**Response:** Thank you for pointing this out. Corrected (Line 261, 307).

**Comment 46:** Line 260: You refer to Fig. S4 before Fig. S3 is used. I think the order of the two supplementary figures need to be changed."

**Response:** Thank you, we have corrected it.

**Comment 47:** Line 279: Which data sets are you referring to? It is unclear."

**Response:** The term "datasets" refers to satellite rainfall datasets that can potentially be used to quantify the uncertainty. We have modified the sentence for clarity. The sentence now reads as follows: "Due to the strong correlation between different microwave satellite-based rainfall datasets — such as IMERG, GSMaP (Kubota et al., 2024), and CMORPH-CDR (Xie et al., 2019) — and the fact that IMERG-Final V7 significantly outperforms other satellite datasets (Wang et al., 2025b), we were unable to quantify the uncertainty arising from the choice of reference data." (Line 280-286)

**Comment 48:** Line 280: Did you consider the random error provided in the IMERG data set?

**Response:** We did not consider the random error provided in the IMERG dataset. This is a great idea, however, it is beyond the scope of the current study, although we may explore using it in a follow-up study.

**Comment 49:** Lines 285-292: It may be worth considering to integrate parts of this paragraph to the conclusion section.

**Response:** Thank you for the suggestion. We have added a line to the conclusions section (Line 403-406).

**Comment 50:** Line 295: Can you really conclude this from your study? Precipitation is the end product of a long chain of processes which can have compensating errors. As pointed out earlier, if you mix both 2D and 3D variables here, this can be misleading.

**Response:** Thank you for this critical comment. We understand the concern regarding potential misinterpretation when combining precipitation with variables of differing dimensionality, which may introduce compensating errors. However, in this section, our analysis is strictly limited to 2D surface observations from METAR stations, including 2-m temperature, 2-m relative humidity,

10-m wind speed, and GPM-IMERG surface precipitation. No 3D atmospheric variables — such as vertical profiles from radiosondes — were included.

**Comment 51:** Line 314-315: "WRF model has indicated….". This sentence sounds a bit awkward.

**Response:** Agreed. After careful reconsideration, we have decided to remove this sentence, as it did not follow naturally from the preceding paragraph, and did not support its arguments.

**Comment 52:** Line 321: "The WRF model…."

**Response:** Thank you for the comment. We have revised the sentence beginning with "The WRF model… (Line 324)"

**Comment 53:** Line 327: Which simplifications? Please elaborate.

**Response:** We have elaborated on the specific simplifications and revised the sentence as follows:

"First, potential deficiencies in the MP, BL, and convection schemes—along with model simplifications such as the absence of data assimilation, limited land surface complexity, simplified radiation and turbulence parameterizations, and the exclusion of aerosol–cloud interactions—may contribute to inaccuracies in simulating moisture convergence and convective updrafts (Taraphdar et al., 2021; Attada et al., 2022). These limitations include simplified representations of land–atmosphere interactions, unresolved sub-grid processes, and the use of prescribed lateral boundary conditions updated every 6 hours, which may not fully capture fast-evolving or small-scale features entering the domain" (Line 329-333)

**Comment 54:** Line 337: "the best performing combination in terms of rainfall".

**Response:** Thank you for your comment. Modified as suggested (Line 343).

**Comment 55:** Figure 6: Showing negative values for accumulated precipitation is not reasonable. Please start with 0 mm or 1 mm.

**Response:** We agree and have adjusted the color scale in Figure 6 to begin at 0 mm, ensuring that only valid, non-negative precipitation values are shown in the figure.

**Comment 56:** Lines 340-344: These sentences are confusing. It is not clear to me what you want so say here, especially in relation to the best performing combination.

**Response:** Thank you for bringing this to our attention. The KGE consists of three components, of which we want to compare the magnitude, so we can make statements regarding the relative importance of each in the final KGE scores. After careful consideration, we have revised this

subsection to enhance the clarity, as we agree it was rather confusing, and we did not answer the main question addressed in this subsection.

**Comment 57:** Figure 7: "… from WRF using the best performing…"

**Response:** Thank you for your suggestion. Changed.

**Comment 58:** Line 345: The part of the sentence in parentheses can be deleted as this is already explained in line 336.

**Response:** Thank you for the suggestion. Done (Line 350).

**Comment 59:** Line 350: "… for the 17 EREs is 0.20 (not shown)…". I also would not call the values of "r-1" scores. In my opinion, it is simply a value.

**Response:** Thank you for your careful observation. We have replaced "scores" with "values" when referring to the harmonized KGE component values.

**Comment 60:** Section 4.9: It this really relevant for your study? My personal feeling is that this is a bit out of scope.

**Response:** Thank you for your comment. We acknowledge your concern regarding the relevance of Section 4.9. However, we have chosen to retain this section, as it provides a synthesis of the key findings in relation to our research objectives and supports the overall flow of the manuscript. We believe it adds value by reinforcing the interpretation of results and guiding the broader implications of our analysis.

**Comment 61:** Line 374: What is a "high-performing scheme"?

**Response:** We have revised the sentence as follows:

"Thus, while several studies employed schemes previously shown to perform well in similar regional contexts, others might have improved simulation accuracy by incorporating the YSU BL scheme and advanced MP schemes identified as effective in our study." (Line 372-374)

**Comment 62:** Section 4.10: Maybe this can be integrated to the conclusion section?

**Response:** We appreciate the comment. We have removed Section 4.10, as per the suggestion by Reviewer 1.

**Comment 63:** Line 389: Please give an example for arid or semi-arid regions with similar characteristics.

**Response:** Thank you for the suggestion. We have removed Section 4.10, as per the suggestion by Reviewer 1.

**Comment 64:** Line 396: Other boundary layer schemes…

**Response:** Revised as follows:

"The YSU (BL1) scheme outperformed the other BL schemes, achieving a temporal KGE of 0.43 and a spatial KGE of 0.29." (Line 382-383)

**Comment 65:** Line 402-407: Did you consider investigating cloud properties like integrated cloud water content (e.g., from CMSAF CLAAS: https://doi.org/10.5676/EUM_SAF_CM/CLAAS/V003)? This can give you a hint of what is happening inside the MP schemes with respect to the cloud formation and thus precipitation.

**Response:** We really appreciate the suggestion. We acknowledge that investigating cloud properties such as integrated cloud water content could offer important insights into the internal behavior of MP schemes, particularly in terms of cloud formation and its relationship to precipitation. However, this study focused on an integrated evaluation using the KGE metric to assess overall model performance across multiple surface variables and precipitation. Nevertheless, we recognize the value of incorporating cloud property diagnostics and will consider this in future work on this topic.

**Comment 66:** Line 423: "This underlines the complexity…". Also radiation has an impact on cloud evolution.

**Response:** We have revised the sentence to reflect this added complexity:

"This underlines the complexity of model parameterization, particularly as cloud evolution is influenced not only by PBL and MP schemes but also by radiative processes, emphasizing the need for further integrated research" (Line 412-414)

**Comment 67:** Line 425: Only for a particular physics combination.

**Response:** We have revised the sentence for clarity as follows:

"For the best-performing physics combination (MP8_BL1), the spatial patterns of simulated and observed rainfall were generally well captured, although occasional overestimations and underestimations were noted. These discrepancies are likely attributable to limitations in the boundary conditions (ERA5 forcing) and uncertainties associated with the IMERG satellite-based reference dataset." (Line 416-419)

**Comment 68:** Line 432: As already mentioned, please reconsider if this part/question is necessary and gives a benefit. This is not a conclusion.

**Response:** Thank you for the comment. We have retained this part and rephrased the question and response to better align with the conclusion. The revised version reads:

"How do the PBL and MP schemes used in previous studies compare with those identified as optimal in our evaluation?

Our findings align with several previous studies in the Middle East that employed the YSU PBL scheme, reinforcing its effectiveness for simulating regional atmospheric dynamics. At the same time, our results suggest that studies using simpler MP schemes—such as Lin (MP2) or Eta Ferrier (MP5)—may achieve improved simulation accuracy by adopting more advanced schemes like Thompson (MP8)." (Line 426-430)

**Comment 69:** Line 440: "Similar climatic conditions…" like?

**Response:** We removed this section as per the suggestion of the reviewer 1.

---

## Author Comment (AC5)

**Pointwise replies to reviewer's comments on the manuscript "Evaluating Microphysics and Boundary Layer Schemes in WRF: Assessment of 36 Scheme Combinations for 17 Major Storms in Saudi Arabia" (egusphere-2025-912)**

**Response to the comments of Reviewer 3**

We thank Dr. Lodh for his thoughtful and constructive feedback. Below are our detailed responses to each comment:

**Comment 1:** Consider making the title slightly more concise and catchier. For example: "Evaluation of Microphysics and Boundary Layer Schemes for Simulating Extreme Rainfall events over Saudi Arabia using WRF".

**Response:** Thank you for the title suggestion. We really like it and have used it. We also incorporated the WRF model version (WRF-ARW v4.4) as suggested by Reviewer 2.

**Comment 2:** In Abstract kindly rephrase the line: "Kling-Gupta Efficiency (KGE) incorporates correlation, variability, and overall bias." to "The Kling-Gupta Efficiency (KGE) metric, which incorporates correlation, variability, and bias, was used for performance evaluation." Provide necessary (original WMO) citations for the metrics used.

**Response:** Thank you for the suggestion. We have revised the sentence to read "The Kling-Gupta Efficiency (KGE), which incorporates correlation, variability, and bias, was used as performance metric." The original citations for the KGE metric (Gupta et al., 2009; Kling et al., 2012) are already provided in the model assessment approach section (Line 7-8, 147-148, 150).

**Comment 3:** In section 1 where you structure the ten key questions, consider using letters (a, b, c...) for questions to avoid confusion with numbered sections.

**Response:** Thank you for this comment. We have replaced the numbered list of key research questions with alphabetically labeled points (a, b, c, ...). However, it is up to the typesetter whether they will adopt this in the final version of the paper.

**Comment 4:** Ensure all acronyms (e.g., MP, BL, KGE, IMERG) are defined on their first use in both the abstract and main text.

**Response:** All acronyms are now defined upon first use in both the abstract and the main text for clarity.

**Comment 5:** Add more region-specific references: While many global references are cited, consider including more recent or specific studies on EREs or WRF performance over Saudi Arabia or the Middle East (e.g., 2022–2024 publications if available).

**Response:** Thank you for the suggestion. The original manuscript already includes recent studies focusing on EREs and WRF performance over Saudi Arabia. Additionally, we will incorporate a few more relevant references covering the broader Middle East region, including publications from 2022 to present (Line 36-37).

Luong, T. M., Dasari, H. P., Attada, R., Chang, H. I., Risanto, C. B., Castro, C. L., ... & Hoteit, I. (2025). Rainfall climatology and predictability over the Kingdom of Saudi Arabia at subseasonal scale. *Quarterly Journal of the Royal Meteorological Society*, e5015.

Taraphdar, S., Gopalakrishnan, D., Liu, C., Pauluis, O. M., Xue, L., Ajayamohan, R. S., ... & Tessendorf, S. A. (2025). Subtropical jet regulates Arabian winter precipitation: A viable mechanism. *Journal of the Atmospheric Sciences*, *82*(4), 713-732.

Francis, D., Fonseca, R., Nelli, N., Cherif, C., Yarragunta, Y., Zittis, G., & Jan de Vries, A. (2025). From cause to consequence: examining the historic April 2024 rainstorm in the United Arab Emirates through the lens of climate change. *npj Climate and Atmospheric Science*, *8*(1), 1-14.

**Comment 6:** On page 2, after line number 25, rephrase the line "These events are often linked to the intrusion of intensified subtropical jet stream…" to: "These events are frequently associated with intrusions of an intensified subtropical jet stream…"

**Response:** Thank you for pointing this out. Done (Line 28-31).

**Comment 7:** Make the figure captions of Figure 2, 3 and 4 more self-explanatory by specifying metrics, datasets, and periods used.

**Response:** Thank you for pointing this out. Captions for Figures 2, 3, and 4 have been revised to provide more detail.

**Comment 8:** In the abstract section after line number 10, "The Thompson-YSU combination yielded the highest mean KGE…" Rephrase to "Among all 36 combinations, the Thompson-YSU pairing consistently produced the highest mean KGE across the 17 storm events.".

**Response:** Revised as suggested (Line 11-12).

**Comment 9:** Ensure consistent use of terms like "EREs," "events," "storms" throughout the paper. Stick with one preferred term unless differentiation is needed.

**Response:** Thank you for the suggestion. We have reviewed the manuscript and ensured consistent use of the term "EREs" throughout the text. This term is now used uniformly unless a different term is specifically required for contextual clarity.

**Comment 10:** After line number 285, "...models often struggle to replicate the spatial distribution of events precisely." Suggestion is to rephrase: "This is expected, as localized convective systems common in the region present challenges for accurately resolving spatial rainfall patterns in mesoscale models."

**Response:** Thank you for the suggestion. We have added a sentence conveying this message in our own words: "This is expected, as accurately simulating the location of localized convective systems remains a major challenge" (Line 293-294).

**Comment 11:** After line number 290: "...the Goddard (MP7) and Thompson (MP8) MP schemes, particularly when paired with the YSU (BL1)…... emerged as superior." Rephrased to "...the Goddard (MP7) and Thompson (MP8) schemes, when combined with YSU (BL1), consistently ranked highest across both temporal and spatial KGE assessments."

**Response:** Thank you for the suggestion. Done (Line 296-297).

**Comment 12:** When discussing major findings (e.g., Thompson–YSU being best), consider referencing the figure or table that supports this claim.

**Response:** Agreed, we have improved our referencing of figures and tables for clarity.

**Comment 13:** The paper can be redrafted to explain the section 4.7 in the beginning i.e. before section 4.1. This is so that readers gets a visual demonstration of the rainfall event in the domain of the study.

**Response:** Thank you for the suggestion. However, Section 4.7 presents the spatial rainfall distribution for the best-performing scheme combination (MP8_BL1), which is identified based on analyses in Sections 4.1 and 4.2. After careful discussion with the co-authors, we believe retaining the current structure preserves the logical flow of the manuscript, and have opted to keep Section 4.7 in its original position.

**Comment 14:** In the conclusion section of the study bring out the motivation/conclusion of the study that this is a kind of a verification study for hydrometeorology.

**Response:** We appreciate the suggestion. The term "verification study for hydrometeorology" has been appropriately incorporated into the conclusion to better reflect the motivation and contribution of the work (Line 379).

**Comment 15:** The authors can also verify the 850hPa wind and near surface temperature and provide plots in supplementary section.

**Response:** As suggested, we have verified the 850 hPa wind and near-surface temperature fields and included the corresponding plots in the supplementary material (Figures S3 and S4).

---

## Referee Report (RR1)

**Review of the revised version of "Evaluation of Microphysics and Boundary Layer Schemes for Simulating Extreme Rainfall Events over Saudi Arabia using WRF-ARW v4.4" by Sahu et al.**

The authors put significant effort in revising the manuscript which is now substantially improved.

However, before the manuscript can be published, I have a few minor points which should be addressed:

1) For Fig. 1 I suggest using a different color table. Depending on the screen and printer of the reader, the reddish/orange colors can hardly be distinguished. I suggest a color table like this: https://www.ncl.ucar.edu/Document/Graphics/ColorTables/topo_15lev.shtml

2) Please briefly mention the advantage of the 2-way nesting approach in the manuscript. Then it is fully clear for the reader what your reasons are.

3) Concerning the application of ERA5 pressure level data: I fully understand that running lots of simulations is computationally expensive, but I clearly see the benefit of using ERA5 model level data for *initialization* of your 17 EREs. The ERA5 data volume for the initial conditions from ERA5 would have increased by a factor of three while there is no change in the data volume of the simulations itself. Please add a brief explanation about potential drawbacks using ERA5 data on pressure levels instead of model levels for model initialization.

4) Please check the grid sizes in Table 3. The numbers are incorrect.

5) Regarding the interpolation of the WRF simulation data to the IMERGE grid: I still see the potential for a double penalty of the model. It is not about improving your results but rather than doing it more realistically in light of the model's ability to represent features which are in the range of 3-6 times the grid distance (see e.g., Skamarock 2004: https://journals.ametsoc.org/view/journals/mwre/132/12/mwr2830.1.xml)

6) I am sorry to mention it again, but in Figure 6+7 there should be no white area for values below zero. I suggest applying a grey background and setting the minimum value to either 0.1 mm or 1.0 mm and then using white colors below 0.1 mm. E.g., 0.1 mm d$^{-1}$ is hard to be measured (there may also be dewfall).

7) Regarding your answer to my comment 65: Keep in mind that you do the surface evaluation with only nine stations over an area of approx. 3000km*2700 km and that the variables you evaluate are diagnostic variables in the model. You may consider the study of Branch et al. (2021) who evaluated WRF simulations (although with an older version of WRF) over the United Arab Emirates (which likely has a similar climate during summer) using ~50 surface stations (https://doi.org/10.5194/gmd-14-1615-2021). Their study shows that a wind bias is apparent in the desert.

---

## Author Response (AR2)

**Author's Response to Editor's Comments**

**Manuscript ID:** EGUSPHERE-2025-912

**Title (revised):** Evaluation of Microphysics and Boundary Layer Schemes for Simulating Extreme Rainfall Events over Saudi Arabia using WRF-ARW.

Dear Prof. Ulbrich,

We thank you for the positive evaluation of the revised manuscript and for summarizing the key remaining points. In what follows, we address all comments from the Editor and Reviewers in detail and have revised the manuscript accordingly. A tracked-changes version has been provided.

**comment 1:** Title: Please leave out the model version number in the title.

**Response:** We agree and have removed the explicit model version number from the title. The revised title now reads:

**"Evaluation of Microphysics and Boundary Layer Schemes for Simulating Extreme Rainfall Events over Saudi Arabia using WRF-ARW"**

This also aligns with Reviewer 1's suggestion.

**Comment 2:** References: Including a large number of references in the introductory text is not beneficial for readers. You should concentrate on a couple of citations which are particularly adequate in the context of the paper.

**Response:** We agree and have substantially streamlined the references in the introduction. In sentences where we previously cited many studies (e.g., statements about the increasing intensity/frequency of extreme rainfall and Clausius–Clapeyron scaling), we now retain only the most relevant and representative references. This improves readability and keeps the focus on the most pertinent literature.

**Comment 3.** Study aims: I agree with the reviewer that your list of study aims/research questions is not the optimum approach. Reading them again, they are mostly not individual questions but belong to each other. For example, a and b refer to two components of the model, and it is not clear per se if they should/can be considered separate from each other. This can be communicated in a text, better than in bullet points. The question of statistical significance may not have to be addressed in a separate bullet point (d). All in all, I am convinced that a short description of the research goals is better suited than the bullet points.

**Response:** The previous list of nine key questions has been removed and concisely integrated into the text. The revised paragraph (in the Introduction, "Study aims" paragraph) reads:

"Our study addresses this gap by conducting an extensive evaluation of WRF-ARW PBL and MP schemes for simulating EREs over the AP at convection-permitting resolution (3~km) to determine the best combination of PBL and MP schemes. We simulate the EREs using a two-way nested domain configuration with 53 vertical levels and horizontal resolutions of 9 and 3~km. We analyze 17 EREs from 2010 to 2022 across the AP, testing 36 different combinations

of PBL and MP schemes. The Kling–Gupta Efficiency (KGE) is used to evaluate the model's performance. We also analyze which component of the KGE exerts the dominant control on the overall KGE scores and whether the performance ranking of schemes is statistically significant and robust across other meteorological variables (2-m air temperature, 2-m relative humidity, and 10-m wind speed). Additionally, we investigate the temporal and spatial consistency of the rainfall evaluation. Lastly, we compare the PBL and MP schemes identified by our assessment as the most effective with those frequently used in previous studies." (Page No. 3, Lines 66-84)

**Comment 4:** With respect to the white areas in Figs. 6 and 7, I think that there is no need for an additional grey color. Rather, it must be clear that there is no values below 0 (i.e. the color bar has to be adapted). You should describe in the captions what "0" means (0.1. mm?) and explain what value is assigned to the white areas.

**Response:** We thank the Editor for this helpful comment. In the revised Figs. 6 and 7, the colour bar has been adapted so that it starts at 0.1 mm and does not include values below 0. We have updated the figure captions to clarify that the white areas indicate grid points with rainfall < 0.1 mm.

We trust these revisions meet your expectations and thank you for guiding us to improve the clarity and quality of our manuscript.

Sincerely,
Dr. Rajesh Kumar Sahu
(on behalf of all co-authors)
KAUST, Saudi Arabia

**Pointwise replies to reviewer's comments on the manuscript "Evaluation of Microphysics and Boundary Layer Schemes for Simulating Extreme Rainfall Events over Saudi Arabia using WRF-ARW" (egusphere-2025-912)**

**Response to the comments of Reviewer 1**

We thank Reviewer 1 for their positive evaluation of the revised manuscript and for the remaining helpful comments, which we have addressed as follows.

**Comment 1:** Although this was proposed by a fellow reviewer, for conciseness, I would remove the version of the model from the title ("-ARW v4.4"). In any case, I leave this decision to the authors.

**Response:** We agree. As also requested by the Editor, we have removed the explicit version number from the title, and now refer simply to "WRF-ARW" (see revised title above).

**Comment 2:** In several instances, mainly in the introduction, some statements are supported by an excessively large number of references (for example: "These events are becoming more frequent and intense as atmospheric moisture increases by about 7% per degree of warming, following Clausius-Clapeyron scaling"). In such cases, citing one or two key references should suffice.

**Response:** We agree and have reduced the number of references in the introduction, keeping only a few key citations for each general statement to make the text more concise and readable.

**Comment 3:** A previous remark suggested presenting the study's aims as a paragraph rather than a list of questions. This aspect remains unchanged, though I continue to believe that a paragraph format would be more appropriate. The same applies to the style of the section headings.

**Response:** We thank the reviewer for this suggestion. We have integrated the questions into a single paragraph summarising the study aims and approach. (Page no. 3, Lines 66-84)

**Comment 4:** In Table 2, under the 'Simulation Start' column, I recommend removing the time indication, since all simulations start at 00:00 (presumably UTC)."

**Response:** We agree. In Table 2, we have removed the redundant time indication in the "Simulation Start" column and now list only the date. The text clarifies that all simulations start at 00:00 UTC, so the table is now cleaner without losing information.

**Response to the comments of Reviewer 2**

We thank Reviewer 2 for the careful re-evaluation and for acknowledging the substantial improvement of the manuscript. We appreciate the remaining detailed minor comments and address them point-by-point below.

**Comment 1:** For Fig. 1 I suggest using a different color table. Depending on the screen and printer of the reader, the reddish/orange colors can hardly be distinguished. I suggest a color table like this: https://www.ncl.ucar.edu/Document/Graphics/ColorTables/topo_15lev.shtml

**Response:** Thank you for this suggestion. We have updated the colour table used in Figure 1 to a multi-level, topography-style colour map inspired by the "topo_15lev" palette. The figure caption has been updated accordingly.

**Comment 2:** Please briefly mention the advantage of the 2-way nesting approach in the manuscript. Then it is fully clear for the reader what your reasons are.

**Response:** We have added a short explanation in the WRF-ARW model configuration section. The text now states that "We used a two-way nesting approach to allow feedback between the high-resolution inner domain and the coarser parent domain. This is essential for capturing small-scale processes like convection, PBL turbulence, and orographic effects, which can influence larger-scale circulation. The dynamic interaction improves physical consistency and is crucial for realistically simulating mesoscale convective systems (MCS) and associated rainfall." (Page no. 6, Lines 134-137)

**Comment 3:** Concerning the application of ERA5 pressure level data: I fully understand that running lots of simulations is computationally expensive, but I clearly see the benefit of using ERA5 model level data for initialization of your 17 EREs. The ERA5 data volume for the initial conditions from ERA5 would have increased by a factor of three while there is no change in the data volume of the simulations itself. Please add a brief explanation about potential drawbacks using ERA5 data on pressure levels instead of model levels for model initialization.

**Response:** We agree this clarification is important. In the data and methods section, we now include a brief discussion:

"ERA5 also provides model-level fields, which offer a finer native representation of the vertical structure, particularly in the boundary layer and near the tropopause. Using model levels for all 17 EREs and 36 parameterization combinations (612 simulations) would substantially increase the initial-condition data volume and I/O burden, particularly when combined with a denser WRF vertical grid, which further increases computational cost. We therefore used pressure levels as a pragmatic compromise, and our results should be interpreted with the caveat that some small-scale vertical features may be under-resolved." (Page no. 5, Lines 114-119)

**Comment 4:** Please check the grid sizes in Table 3. The numbers are incorrect.

**Response:** Thank you for pointing this out. We have re-checked and corrected the grid spacing values in Table 3.

**Comment 5:** Regarding the interpolation of the WRF simulation data to the IMERGE grid: I still see the potential for a double penalty of the model. It is not about improving your results but rather than doing it more realistically in light of the model's ability to represent features which are in the range of 3-6 times the grid distance (see e.g., Skamarock 2004: https://journals.ametsoc.org/view/journals/mwre/132/12/mwr2830.1.xml)

**Response:** We appreciate this comment. As described in the model assessment approach, WRF-ARW rainfall at 3-km resolution is resampled to the 0.1° IMERG grid using averaging, which is more than three times the grid spacing (consistent with Skamarock, 2004), to enable consistent grid-cell-to-grid-cell comparison and largely avoid the double-penalty issue.

**Comment 6:** I suggest applying a grey background and setting the minimum value to either 0.1 mm or 1.0 mm and then using white colors below 0.1 mm. E.g., 0.1 mm d-1 is hard to be measured (there may also be dewfall).

**Response:** We appreciate the reviewer's comment. In the revised Figs. 6 and 7, the colour bar has been adapted so that it starts at 0.1 mm and does not include values below 0. We have updated the figure captions to clarify that the white areas indicate grid points with rainfall < 0.1 mm.

To keep the figures visually simple and consistent with the Editor's comment, we did not introduce a grey background, but we believe the explicit caption now fully prevents misinterpretation.

**Comment 7:** Regarding your answer to my comment 65: Keep in mind that you do the surface evaluation with only nine stations over an area of approx. 3000km*2700 km and that the variables you evaluate are diagnostic variables in the model. You may consider the study of Branch et al. (2021) who evaluated WRF simulations (although with an older version of WRF) over the United Arab Emirates (which likely has a similar climate during summer) using ~50 surface stations (https://doi.org/10.5194/gmd-14-1615-2021). Their study shows that a wind bias is apparent in the desert.

**Response:** We thank the reviewer for this helpful comment and for bringing Branch et al. (2021) to our attention. We fully agree that a dense surface network and a longer-term evaluation of diagnostic near-surface variables are highly valuable. However, our study is explicitly designed around 17 selected EREs, so both the WRF simulations and the verification strategy are event-focused, and reliable concurrent observations are only available at nine stations near these events. Note that most of the Arabian Peninsula is very sparsely observed, with only a few dozen unevenly distributed gauges over a vast area, unlike the UAE.